# KLK3/PSA and cathepsin D activate VEGF-C and VEGF-D

**Sawan Kumar Jha[1,2†], Khushbu Rauniyar[1†], Ewa Chronowska[1,3], Kenny Mattonet[4], Eunice Wairimu Maina[1], Hannu Koistinen[5,6], Ulf-Håkan Stenman[5,6], Kari Alitalo[2,6,7], Michael Jeltsch[1,2]\***

[1]Individualized Drug Therapy Research Program, University of Helsinki, Helsinki, Finland; [2]Wihuri Research Institute, Helsinki, Finland; [3]Jagiellonian University Medical College, Cracow, Poland; [4]Max Planck Institute for Heart and Lung Research, Bad Nauheim, Germany; [5]Department of Clinical Chemistry, University of Helsinki, Helsinki, Finland; [6]Helsinki University Hospital, Helsinki, Finland; [7]Translational Cancer Medicine Research Program, University of Helsinki, Helsinki, Finland

**\*For correspondence:**
michael@jeltsch.org

[†]These authors contributed equally to this work

**Competing interests:** The authors declare that no competing interests exist.

**Abstract** Vascular endothelial growth factor-C (VEGF-C) acts primarily on endothelial cells, but also on non-vascular targets, for example in the CNS and immune system. Here we describe a novel, unique VEGF-C form in the human reproductive system produced via cleavage by kallikrein-related peptidase 3 (KLK3), aka prostate-specific antigen (PSA). KLK3 activated VEGF-C specifically and efficiently through cleavage at a novel N-terminal site. We detected VEGF-C in seminal plasma, and sperm liquefaction occurred concurrently with VEGF-C activation, which was enhanced by collagen and calcium binding EGF domains 1 (CCBE1). After plasmin and ADAMTS3, KLK3 is the third protease shown to activate VEGF-C. Since differently activated VEGF-Cs are characterized by successively shorter N-terminal helices, we created an even shorter hypothetical form, which showed preferential binding to VEGFR-3. Using mass spectrometric analysis of the isolated VEGF-C-cleaving activity from human saliva, we identified cathepsin D as a protease that can activate VEGF-C as well as VEGF-D.
DOI: https://doi.org/10.7554/eLife.44478.001

## Introduction

Vascular endothelial growth factor VEGF-A is essential for early embryonic development and for successful implantation of the embryo into the uterus (*Binder et al., 2014*). VEGF-A acts in this function on both vascular and non-vascular targets (*Hannan et al., 2011*). The primary function of the closely related growth factor VEGF-C is stimulation of growth of the lymphatic vasculature (*Rauniyar et al., 2018*). VEGF-C is required for ovarian follicle growth and maturation and endometrial lymphangiogenesis (*Rogers, 2008*; *Rutkowski et al., 2013*). Unlike VEGF-A, which is secreted as an active growth factor (*Leung et al., 1989*), VEGF-C is secreted as an inactive precursor (pro-VEGF-C), which requires two proteolytic cleavages for activation (*Jeltsch et al., 2014*; *Joukov et al., 1997*). The first C-terminal cleavage resulting in pro-VEGF-C occurs constitutively in the endoplasmic reticulum and is mediated by proprotein convertases (*Siegfried et al., 2003*). The second cleavage takes place in the extracellular environment, is highly regulated and requires the assembly of a trimeric complex consisting of VEGF-C, the ADAMTS3 metalloproteinase and the 'cofactor' CCBE1 (*Bui et al., 2016*; *Jeltsch et al., 2014*). Alternative activation by plasmin has been shown in vitro, but its significance under physiological settings is unknown (*McColl et al., 2003*). VEGF-D is the closest paralog of VEGF-C (*Achen et al., 1998*). Similar to VEGF-C, it is lymphangiogenic (*Veikkola et al., 2001*), but appears to have a higher angiogenic potential than VEGF-C (*Byzova et al., 2002*; *Rissanen et al.,*

**eLife digest** The lymphatic system is composed of networks of vessels that drain fluids from the body's tissues and filter it back into the blood. Growing these vessels depends on a factor known as VEGF-C, which is released in an inactive form and must be cut by enzymes before it can work. One enzyme that is known to activate the VEGF-C signal when the early embryo is developing is ADAMTS3. If this signal fails to switch on this can result in a condition known as lymphedema – whereby problems in the lymphatic system cause tissues to swell due to insufficient drainage. However, it is unknown whether the VEGF-C signal can be activated by enzymes other than ADAMTS3.

To investigate this Jha, Rauniyar et al. tested a specific family of proteins commonly found in the human prostate, which have previously been predicted to act on VEGF-C. This revealed that the lymphatic vessel growth factor can also be activated by an enzyme found in seminal fluid called prostate specific antigen, or PSA for short. To see if enzymes in other bodily fluids could switch on VEGF-C, different components of human saliva were separated and tested to see which could cut inactive VEGF-C. This showed that VEGF-C could be converted to an active form by another enzyme called cathepsin D.

Unexpectedly, Jha, Rauniyar et al. found that VEGF-C was also present in semen. For conception to occur PSA must liquify the semen following ejaculation. It was discovered that PSA activates VEGF-C just as the semen starts to liquify, suggesting that the lymphatic vessel growth factor might also play an important role in reproduction. In addition to VEGF-C, both PSA and cathepsin D were found to activate another growth factor called VEGF-D, which has an unknown role in the human body.

VEGF-C helps the spread of tumors, and blocking the two enzymes that activate this growth factor may be a new therapeutic approach for cancer. However, more work is needed to validate which types of tumor, if any, use these enzymes to activate VEGF-C. In addition, understanding the relationship between PSA and VEGF-C could help improve our knowledge of human reproduction.
DOI: https://doi.org/10.7554/eLife.44478.002

2003). The proteolytic activation of VEGF-D is very similar to that of VEGF-C (*Stacker et al., 1999a*), but it deploys distinct, so far unknown proteases (*Bui et al., 2016*).

Many kallikrein-related peptidases are highly expressed in the prostate, and some prostate-derived cell lines, such as the immortalized human normal prostate epithelial (NPrEC) or PC-3 cells — from which VEGF-C was originally cloned — express high amounts of VEGF-C (*Grennan, 2006*; *Joukov et al., 1996*). In a peptide library scan, *Matsumura et al. (2005)* identified VEGF-C as a potential substrate for KLK4. Based on these observations, we tested human kallikrein-related peptidases for their ability to activate VEGF-C. In this study, we show that KLK3, the major protease in human semen, is able to specifically activate VEGF-C and VEGF-D. We further show that cathepsin D cleavage of VEGF-C results in a novel, predominantly VEGFR-3-binding form of VEGF-C, and that cathepsin D cleavage of VEGF-D at the homologous site results in a VEGFR-2-specific (minor mature) form of VEGF-D.

## Results

### VEGF-C is processed by the kallikrein-related peptidase 3 (KLK3)

We could not demonstrate robust VEGF-C activation by KLK4 as predicted by *Matsumura et al. (2005)* (data not shown), but purified KLK3 cleaved pro-VEGF-C, resulting in a mature protein that migrated at about 20 kDa in Western blotting analysis (*Figure 1A*, lane 2). To confirm that KLK3 was responsible for the cleavage, we inhibited its protease activity by using the monoclonal antibody 5C7 (*Stenman et al., 1999*) in 2-fold molar excess (*Figure 1A*, lane 1 versus lane 2). We probed the polypeptide bands resulting from the cleavage with rabbit antiserum 6 and antiserum 3/4, which were raised against full-length and mature VEGF-C, respectively (*Figure 1A and B*, compare the second lanes). Probing with antiserum 3/4, which recognizes both pro-VEGF-C and mature VEGF-C, showed that the majority of pro-VEGF-C had been cleaved by KLK3.

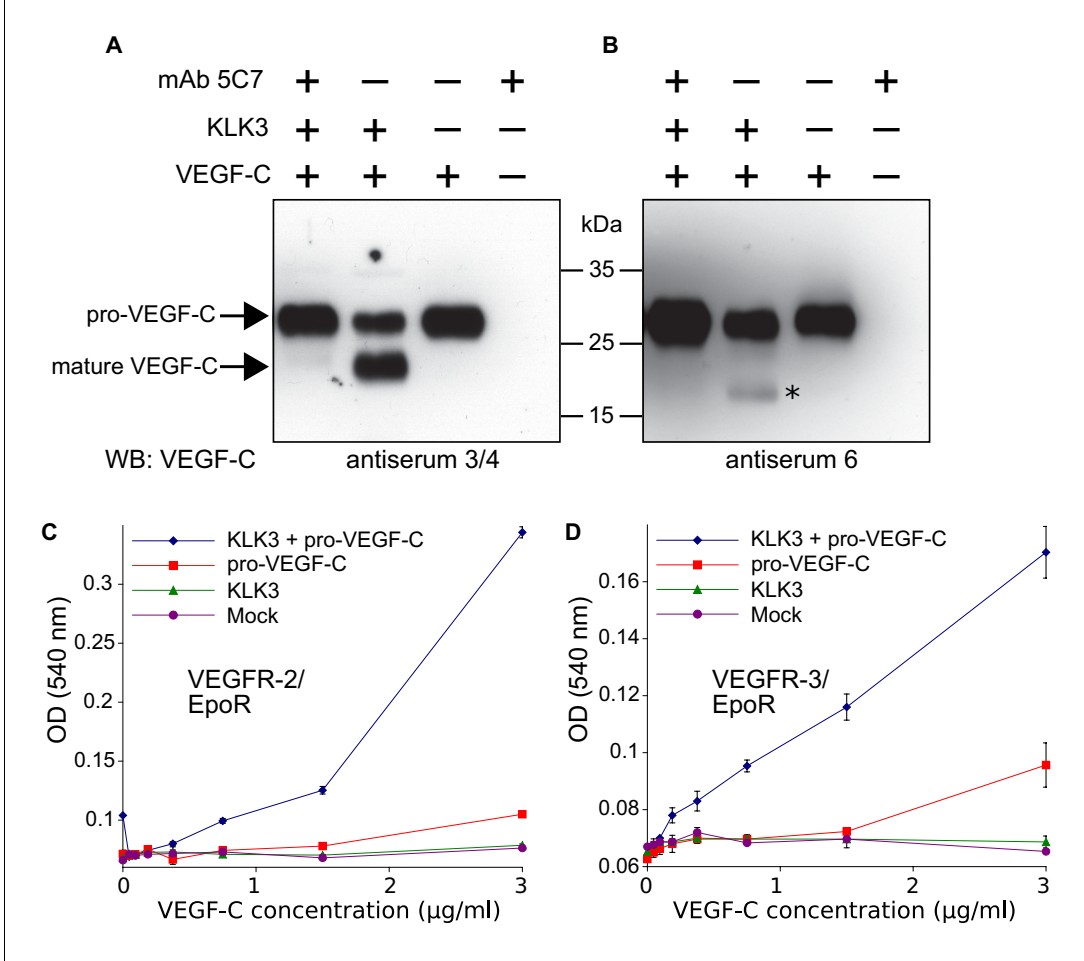

**Figure 1.** Kallikrein-related peptidase 3 (KLK3)/Prostate specific antigen (PSA) activates VEGF-C. (A, B) Cleavage of pro-VEGF-C by KLK3 (PSA). Pro-VEGF-C was incubated with or without KLK3, with and without the monoclonal antibody against KLK3 (5C7). Detection of VEGF-C in Western blots probed with antiserum 6 and 3/4, resulting in the detection of pro-VEGF-C (29/31 kDa) and activated, mature VEGF-C (21/23 kDa). The band marked by the asterisk likely represents the N-terminal propeptide (~15 kDa) which is detected by the antiserum 6. Note that for the image shown for antiserum 6, two different exposures of the same blot were merged (n = 3). (C, D) VEGF-C processed by KLK3 is biologically active in Ba/F3 cell assays, which translate activation of a hybrid VEGFR/EpoR receptor into cell survival (n = 2). Error bars indicate SD.

DOI: https://doi.org/10.7554/eLife.44478.003

The following source data is available for figure 1:

**Source data 1.** Ba/F3 assay showing the activity of KLK3-cleaved VEGF-C.

DOI: https://doi.org/10.7554/eLife.44478.004

## KLK3-processed VEGF-C is biologically active

We tested the KLK3-processed VEGF-C for its biological activity in Ba/F3 cells, which had been stably transfected with VEGFR/EpoR chimeras and found that it promoted the survival of both VEGFR-2/EpoR (*Figure 1C*) and VEGFR-3/EpoR cells (*Figure 1D*).

## KLK3 activation of VEGF-C results in a unique VEGF-C species

Edman degradation of the KLK3-processed VEGF-C revealed the amino-terminal sequence NTEIL (*Figure 2—figure supplement 1*). Thus, KLK3 cleaves VEGF-C between Tyr-114 and Asn-115, targeting a sequence similar to most of its cleavage sites in the seminogelins, which are the primary target proteins of KLK3 (*Malm et al., 2000*). The KLK3-cleaved VEGF-C is three N-terminal amino acid residues shorter than the mature VEGF-C generated by ADAMTS3 (*Jeltsch et al., 2014*) and 12 amino acid residues shorter than the mature VEGF-C produced by PC-3 cells (*Figure 2*)

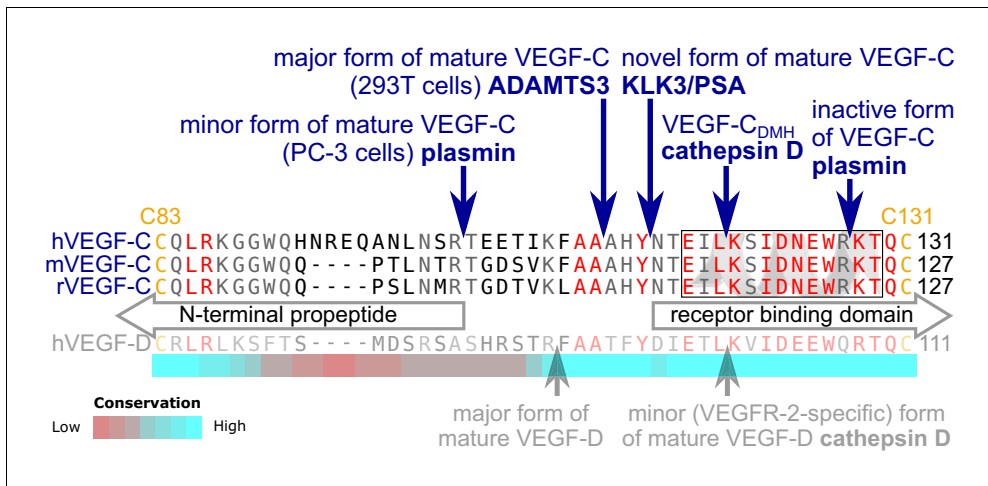

**Figure 2.** KLK3 activation of VEGF-C results in a unique VEGF-C species. KLK3 cleavage results in a mature VEGF-C species that is N-terminally three amino acids shorter than the ADAMTS3-cleaved VEGF-C. Shown are the aligned amino acid sequences between the N-terminal propeptide and the receptor binding domain of VEGF-C in human (h), mouse (m) and rat (r)VEGF-C and hVEGF-D. The arrows mark the sites of proteolytic cleavage of all four reported VEGF-C-activating enzymes and the two reported cleavage sites in VEGF-D. Residues within the N-terminal alpha-helix of VEGF-C/D are boxed. Note that the 1st plasmin cleavage site was not verified experimentally, but deduced from the plasmin cleavage signature and the size of the resulting product. The heat map under the alignment indicates the areas of highest divergence, deduced from a more comprehensive alignment of VEGF-C orthologs (see *Figure 2—figure supplement 2*).
DOI: https://doi.org/10.7554/eLife.44478.005

The following figure supplements are available for figure 2:

**Figure supplement 1.** Results of the Edman degradation of KLK3-cleaved pro-VEGF-C.
DOI: https://doi.org/10.7554/eLife.44478.006
**Figure supplement 2.** Interspecies analysis of VEGF-C amino acid sequences relevant for activation.
DOI: https://doi.org/10.7554/eLife.44478.007

---

(*Joukov et al., 1997*). We analyzed the VEGF-C amino acid sequences of 40 vertebrate species and found that residues −7 to +1 relative to the KLK3 cleavage site and −4 to +4 relative to the ADAMTS3 cleavage site (KFAA↓AHY↓N) are 100% conserved among all mammals and birds that were included in the analysis. However, we found significant differences in this area in all fish species analyzed (*Figure 2—figure supplement 2*).

## Human seminal fluid contains VEGF-C

To evaluate the biological significance of VEGF-C activation by KLK3, we first analyzed the VEGF-C content of human seminal plasma. Because of difficulties in detecting VEGF-C at low ng/ml-range concentrations in a high-protein sample (~50 mg/ml), such as semen (*Owen and Katz, 2005*), we first compared the ability of different anti-VEGF-C antibodies to detect VEGF-C (*Figure 3—figure supplement 1*, *Supplementary file 1*). VEGF-C was detected in Western blots of sperm liquefied for approximately 20–30 min at room temperature by using antibody sc-374628 after VEGF-C precipitation with soluble forms of its receptors VEGFR-2 (VEGFR-2/Fc) and VEGFR-3 (VEGFR-3/Fc) or anti-VEGF-C antiserum 882 (*Figure 3A*). The affinity of seminal plasma VEGF-C towards VEGFR-2 appeared to be much weaker than towards VEGFR-3 in the VEGF-C pull down assay (*Figure 3A*, compare lanes 4 and 6). The mobilities of the VEGF-C polypeptides indicated that it is composed of inactive pro-VEGF-C and active mature VEGF-C. Stimulation of VEGFR-3-transfected porcine aortic endothelial (PAE) cells with seminal plasma resulted in VEGFR-3 phosphorylation (*Figure 3B*, compare lanes 2 and 4). VEGF-C stimulation of PAE cells stably expressing VEGFR-2 led to an even stronger phosphorylation than the recombinant VEGF-C control. We reasoned that this could indicate the presence of VEGF-A, whose concentrations in seminal plasma have been reported to range from less than 2 ng/ml to more than 100 ng/ml (*Obermair et al., 1999*). Indeed, most of the

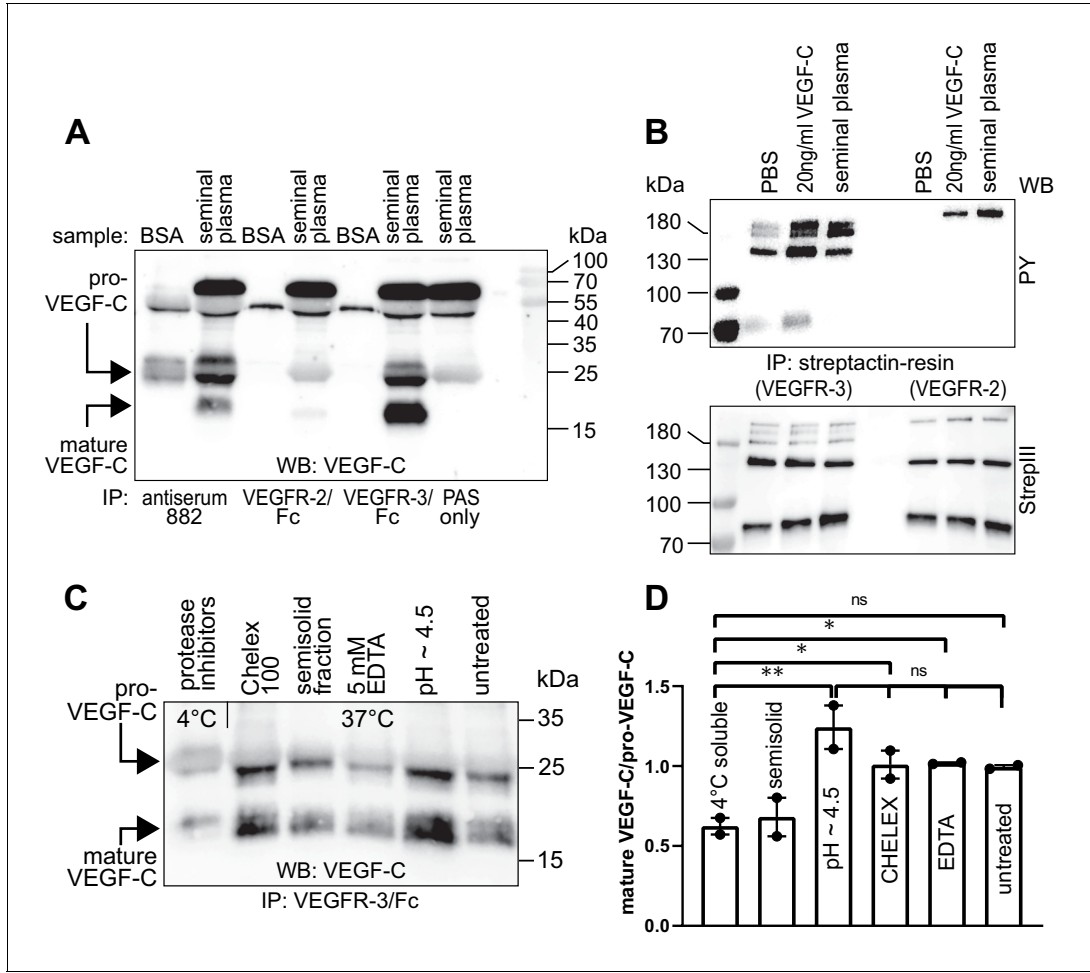

**Figure 3.** Seminal plasma VEGF-C is cleaved during sperm liquefaction and binds to and activates VEGFR-3. (**A**) Detection of both pro-VEGF-C and activated, mature VEGF-C by Western blotting with anti-VEGF-C antibody sc-374628 after pull-down with soluble VEGF receptors or antiserum 882. (**B**) Phosphorylation of VEGFR-2 and VEGFR-3 by seminal VEGF-C. Note that the phosphorylation pattern of VEGFR-3 is slightly different from that induced by 20 ng/ml of the VEGF-C control protein, which corresponds to the plasmin-activated form of VEGF-C. The lower panel shows the same blot reprobed with Streptactin-HRP for detection of the total levels of VEGFR-3 and VEGFR-2, respectively. (PAS, protein A sepharose; PY, phosphotyrosine) (**C**) Fresh seminal fluid contains less processed VEGF-C than seminal fluid liquefied at 37°C, indicating that pro-VEGF-C is converted into mature VEGF-C after ejaculation. Effect of protease inhibitors and low temperature on cleavage of VEGF-C (lane 1). While the mature VEGF-C/pro-VEGF-C ratios of ion sequestered samples (50 mg/ml CHELEX 100 and 5 mM EDTA in lanes 2 and 4, respectively) were not different from the untreated sample (lane 6), lowering the pH tended to increase the activation of VEGF-C (lane 5), but the difference to untreated sample did not reach statistical significance. Note that non-liquefied and liquefied samples differ because the semisolid seminogelins largely disappear during liquefaction (*Malm et al., 2000*). The semisolid fraction of fresh ejaculate was separately assessed for its VEGF-C content after liquefaction (lane 3). (**D**) Quantification of the ratio of mature VEGF-C to pro-VEGF-C in seminal plasma exposed to different conditions. Comparison of the 4°C sample to pH ~4.5 (p=0.0066), CHELEX (p=0.045), untreated (p=0.052), and EDTA (p=0.042) [One-way ANOVA, Dunnett's multiple comparisons test (n = 2), data are presented as mean ± SEM].

DOI: https://doi.org/10.7554/eLife.44478.008

The following source data and figure supplements are available for figure 3:

**Source data 1.** Quantification of the ratio of mature VEGF-C to pro-VEGF-C in seminal plasma.

DOI: https://doi.org/10.7554/eLife.44478.012

**Figure supplement 1.** Comparison of 17 different antibodies for the detection of mature and pro-VEGF-C by Western blotting.

DOI: https://doi.org/10.7554/eLife.44478.009

**Figure supplement 2.** The VEGFR-2 phosphorylating activity of seminal plasma is blocked by the VEGF-A-capturing VEGFR-2/Fc fusion protein.

DOI: https://doi.org/10.7554/eLife.44478.010

**Figure supplement 3.** No detection of VEGF-D in seminal plasma.

DOI: https://doi.org/10.7554/eLife.44478.011

VEGFR-2 phosphorylation was blocked when incubated with soluble VEGFR-2/Fc, but not by incubation with VEGFR-3/Fc (*Figure 3—figure supplement 2*). In contrast, VEGF-D was not detected in seminal plasma (*Figure 3—figure supplement 3*).

## VEGF-C is activated during sperm liquefaction

When fresh ejaculates were immediately mixed with protease inhibitors, placed on ice and analyzed, less active VEGF-C was detected than in ejaculates that had been liquefied, indicating that pro-VEGF-C is converted into mature VEGF-C after ejaculation (*Figure 3C*), concurrently with sperm liquefaction. Lowering the pH with citric acid tended to increase slightly the yield of mature VEGF-C (*Figure 3C*, lane 5), but ion chelation with CHELEX 100 or EDTA had no effect (*Figure 3C*, lanes 2 and 4, respectively).

## VEGF-C processing by KLK3 is enhanced by CCBE1

We have shown that CCBE1 enhances the proteolytic activation of VEGF-C by ADAMTS3, but not by plasmin (*Jeltsch et al., 2014*). Therefore, we tested whether CCBE1 would accelerate KLK3 activation of VEGF-C. We found that KLK3-mediated cleavage of VEGF-C was enhanced by CCBE1 when CCBE1 or KLK3 amounts were titrated down so that only little VEGF-C processing occurred (*Figure 4*). Substantial amounts of CCBE1 were detected in seminal plasma by Western blotting

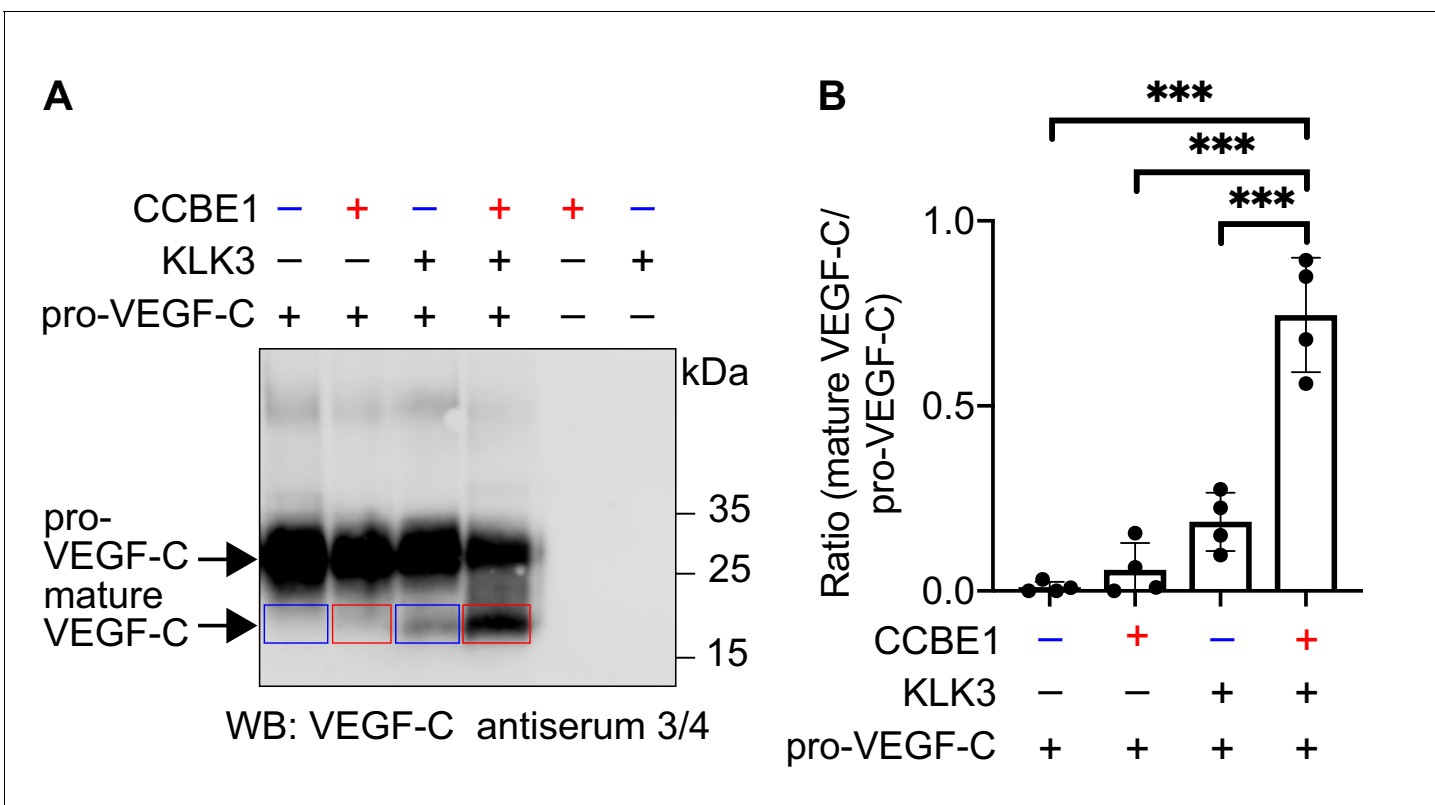

**Figure 4.** VEGF-C activation by KLK3 is enhanced by CCBE1. (**A**) Activation of VEGF-C by KLK3 is enhanced by CCBE1. The mature VEGF-C produced in the presence or absence of CCBE1 is shown in the red and blue boxes, respectively. (**B**) Quantification of the mature VEGF-C/pro-VEGF-C ratio. Data are shown as mean ± SD (n = 4). Statistical differences were determined by one-way ANOVA with Tukey *post hoc* test, ***p<0.001.
DOI: https://doi.org/10.7554/eLife.44478.013

The following source data and figure supplement are available for figure 4:

**Source data 1.** Quantification of the ratio of mature VEGF-C to pro-VEGF-C to show that activation of VEGF-C by KLK3 is enhanced by CCBE1.
DOI: https://doi.org/10.7554/eLife.44478.015

**Figure supplement 1.** Seminal plasma contains CCBE1 protein.
DOI: https://doi.org/10.7554/eLife.44478.014

(*Figure 4—figure supplement 1*), confirming published proteomics results (*Jodar et al., 2016*). This indicates that VEGF-C cleavage could be increased by CCBE1 also in semen.

## VEGF-C and VEGF-D activities have different sensitivities to N-terminal truncations

Activated VEGF-C binds to VEGFR-2 (*Joukov et al., 1997*), but in our assays with seminal plasma, VEGFR-2 binding was very weak (*Figure 3A*, lane 4). To explain this finding, we focused on the cleavage of the N-terminal helix, because its partial removal in VEGF-D decreases selectively VEGFR-3 binding while leaving VEGFR-2 binding intact (*Leppänen et al., 2011*). Since complete proteolytic removal of the N-terminal helix of VEGF-C abolishes all receptor binding and phosphorylation-stimulating activity (*Jeltsch et al., 2014*), we first tested if a partial removal of the N-terminal helix (cutting between Leu-118 and Lys-119, corresponding to the proteolytic cleavage site between Leu-114 and Lys-115 of VEGF-D) would result in a selective loss of VEGF-C binding to its receptors.

The protease that cleaves between Leu-114 and Lys-115 of VEGF-D (and hypothetically between the homologous Leu-118 and Lys-119 of VEGF-C) is unknown. Therefore, we generated this form of VEGF-C by truncating the VEGF-C cDNA, which was then expressed in S2 cells. Interestingly, unlike the corresponding VEGF-D form, this 'VEGF-C$_{DMH}$' (for 'D Minor Homology') bound to VEGFR-3, but only weakly or not at all to VEGFR-2 (*Figure 5A*). Because of this unexpected result, we performed the experiment using proteins produced in 293 T cells and found that in conditions where all other mature forms of VEGF-C interacted with their receptors as predicted, VEGF-C$_{DMH}$ did not bind to VEGFR-2 or VEGFR-3 (*Figure 5B*). Since the loss of binding compared to the S2 cell-produced VEGF-C$_{DMH}$ is not associated with a loss of receptor-interacting amino acid residues, we attributed the loss of binding to the extra N-terminal four amino acid residues of the mammalian linker (see also *Figure 5—figure supplement 1*).

We then used equimolar amounts of truncated VEGF-Cs expressed in transiently transfected CHO cells (*Figure 5—figure supplement 1*) to test the bioactivity of different N-terminally truncated VEGF-Cs in VEGFR-2 and VEGFR-3 phosphorylation assays. The receptor phosphorylation results mirrored the binding results. The longest mature VEGF-C resulted in the strongest stimulation, and progressive shortening of the N-terminus resulted in gradually decreased stimulation of the receptor phosphorylation (*Figure 5C*). We also tested the activity of purified VEGF-C$_{DMH}$ expressed in S2 cells. In agreement with the binding results, 100 ng/ml VEGF-C$_{DMH}$ did stimulate the phosphorylation of VEGFR-3 but not or only very weakly of VEGFR-2 (*Figure 5D*).

A comparison of the sizes of VEGF-C polypeptides produced by S2 cells transfected with N-terminally truncated cDNAs encoding the polypeptide resulting from cleavage by ADAMTS3 and the (longer) form generated by the 1st plasmin cleavage revealed bands of identical size, indicating additional proteolytic processing (*Figure 5—figure supplement 1*). N-terminal sequencing of the form produced from the longer cDNA revealed that about ⅔ had the KSIDNE... N-terminus, and about ⅓ had AAAHYN... as N-terminus. Hence, the DMH-form of mature VEGF-C can also be produced by proteolytic processing of a longer mature form of VEGF-C by a yet unknown protease. We refer to such cleavage on top of an existing activation in the following as *secondary activation* (irrespectively of the receptor activation ability of the resulting protein species).

## Cathepsin D activates both VEGF-C and VEGF-D

The presence of a VEGF-C-cleaving protease in seminal fluid prompted us to search for such a protease also in other body fluids. We enriched the VEGF-C activating component of human saliva by cation exchange chromatography (*Figure 6—figure supplement 1*) and subjected the fractions containing the peak activity to mass spectrometric analysis. Among the highest scoring proteases (*Supplementary file 2*), cathepsin D was identified as the most likely candidate due to the cleavage context of the DMH-form of VEGF-C (Leu-118↓Lys-119). Using purified recombinant proteins, we confirmed that cathepsin D cleaves pro-VEGF-C into active VEGF-C (*Figure 6A* and *Figure 7AB*) and performs a *secondary activation* of the minor, mature form of VEGF-C (*Figure 6A* and *Figure 7F*). Because the sequence contexts of the cleavage sites of cathepsin D and KLK3 are conserved between VEGF-C and VEGF-D (see *Figure 2*), we investigated, if also cathepsin D and KLK3 could activate pro-VEGF-D. Indeed, cathepsin D activated pro-VEGF-D and performed a *secondary activation* of the longer, mature form of VEGF-D. The cleavage of mature VEGF-D was rapid and

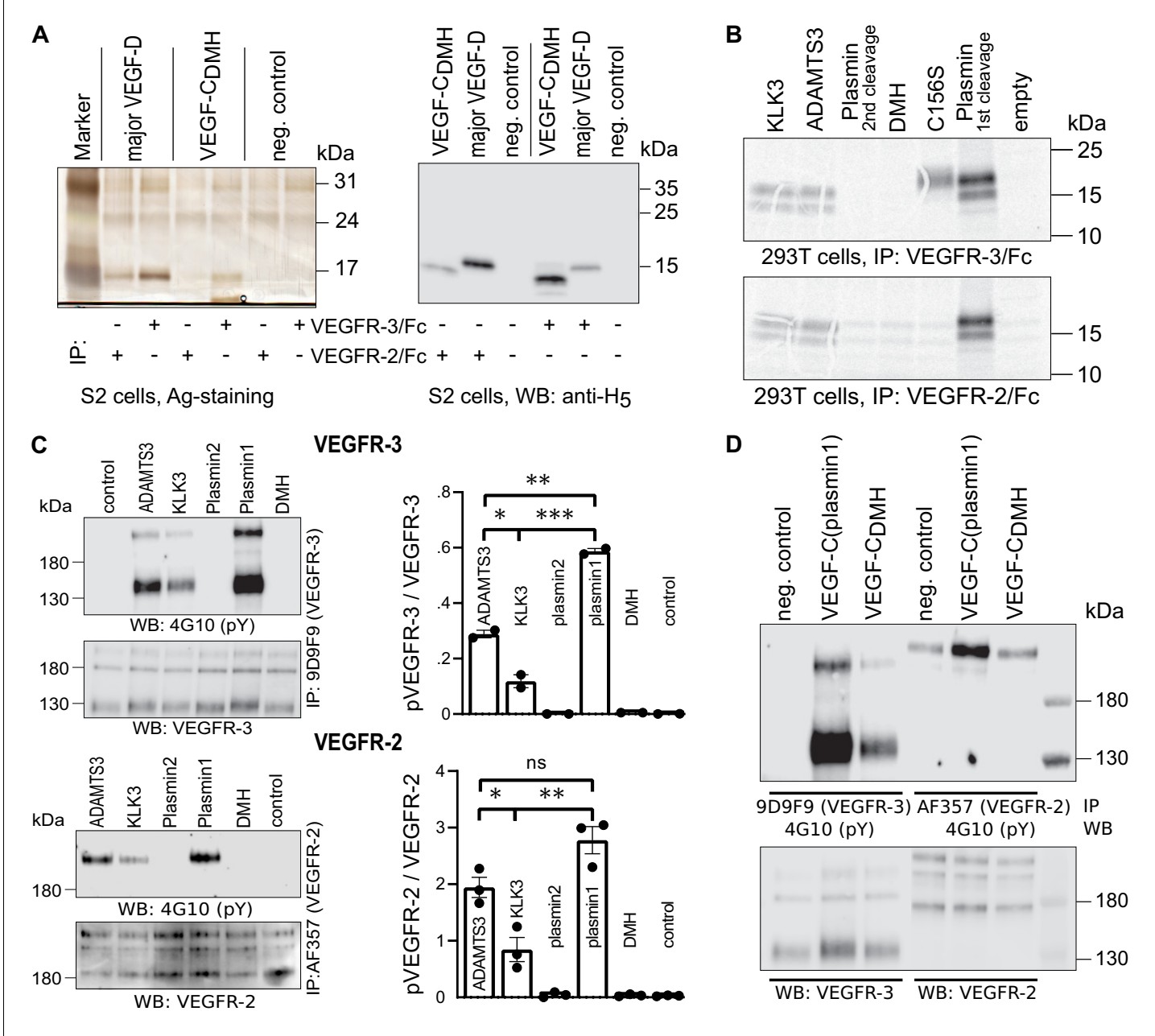

**Figure 5.** Shortening of the VEGF-C N-terminal helix reduces receptor binding and activation. (**A**) VEGF-C$_{DMH}$ form binds efficiently to VEGFR-3 but weakly to VEGFR-2 when expressed in S2 cells. (**B**) Lack of binding of 293T-produced VEGF-C$_{DMH}$ to VEGFR-2 or VEGFR-3. Note that the weak bands visible in the mock transfected 293T samples are due to endogenous VEGF-A, which binds to VEGFR-2, but not to VEGFR-3 (n = 2). (**C**) Stimulation of VEGFR-3 and VEGFR-2 phosphorylation by equimolar amounts of N-terminally truncated VEGF-Cs (corresponding to mature VEGF-C generated by ADAMTS3, KLK3, and the first plasmin cleavage) expressed in CHO cells (quantification: n = 2 for VEGFR-3; n = 3 for VEGFR-2; data are presented as mean ± SEM; one-way ANOVA, Tukey's multiple comparisons test). When compared with control, all three mature VEGF-C forms showed significant stimulation of both receptors (p-values=0.0094 to<0.0001). (**D**) Phosphorylation of hVEGFR-3 but not hVEGFR-2 in PAE cells by VEGF-C$_{DMH}$ purified from S2 cells.

DOI: https://doi.org/10.7554/eLife.44478.016

The following source data and figure supplement are available for figure 5:

**Source data 1.** Quantification of VEGFR-3 and VEGFR-2 receptor phosphorylation by N-terminally truncated VEGF-Cs.
DOI: https://doi.org/10.7554/eLife.44478.018
**Figure supplement 1.** Secondary activation of a longer mature VEGF-C form in S2 cells.
DOI: https://doi.org/10.7554/eLife.44478.017

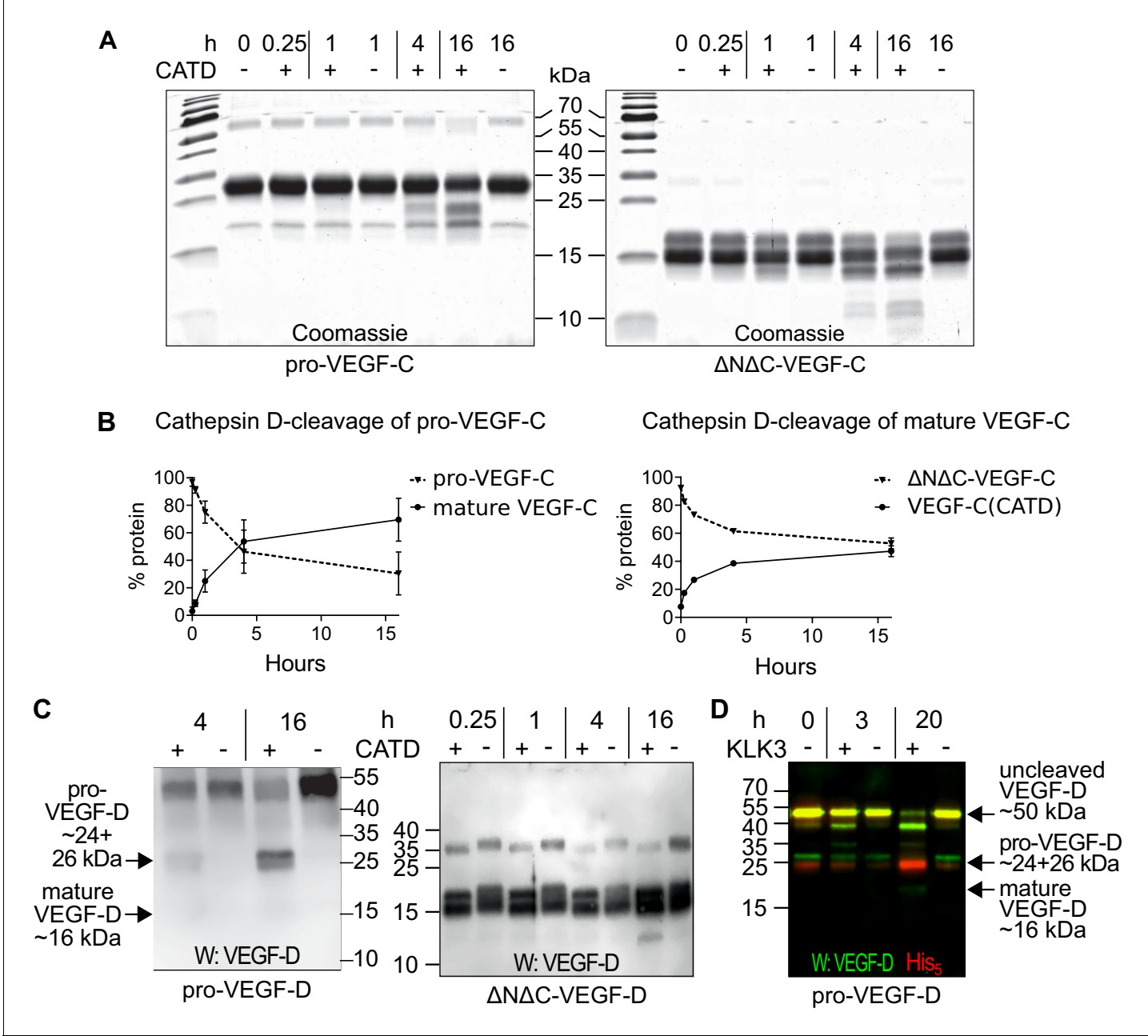

**Figure 6.** Cathepsin D activates pro-VEGF-C/D and mature VEGF-C/D, respectively. (**A**) Cleavage of VEGF-C by coincubation of pro-VEGF-C with cathepsin D (left panel) and secondary activation of ΔNΔC-VEGF-C (a mature form of VEGF-C translated from a truncated cDNA, right panel). (**B**) Quantification of the cleavage of pro-VEGF-C (n = 3) and ΔNΔC-VEGF-C (n = 2) by cathepsin D. (**C**) Cathepsin-D-mediated conversion of pro-VEGF-D into mature VEGF-D (left panel), and rapid activation of ΔNΔC-VEGF-D (a mature form of VEGF-D translated from a truncated cDNA, right panel). (**D**) Cleavage of pro-VEGF-D by KLK3. Note that, KLK3 cleaves VEGF-D between the VEGF homology domain and the N-terminal propeptide, but also between the VEGF homology domain and the C-terminal propeptide (for a detailed breakdown of the cleavage products visible in this overlay and the individual exposures, see *Figure 7—figure supplement 2*) (n = 2).

DOI: https://doi.org/10.7554/eLife.44478.019

The following source data and figure supplement are available for figure 6:

**Source data 1.** Quantification of the cleavage of pro-VEGF-C and mature VEGF-C by cathepsin D.
DOI: https://doi.org/10.7554/eLife.44478.021

**Figure supplement 1.** Enrichment and fractionation of VEGF-C cleaving activity.
DOI: https://doi.org/10.7554/eLife.44478.020

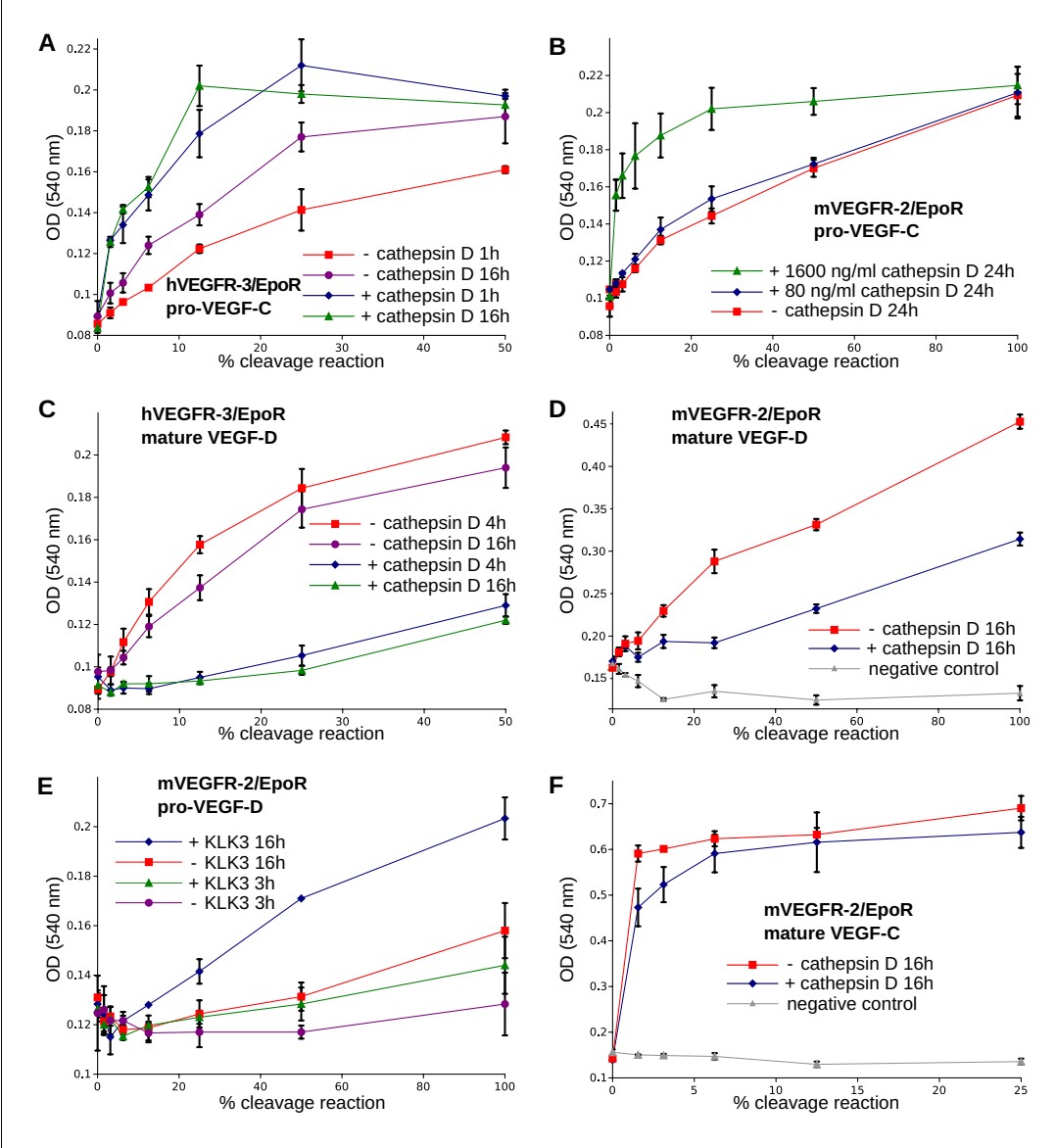

**Figure 7.** The receptor-activating properties of VEGF-C and VEGF-D are differentially affected by cathepsin D cleavage. Shown are the results of Ba/F3-VEGFR/EpoR assays used to evaluate the receptor-activating properties of cathepsin D- and KLK3- cleaved proteins. Cathepsin D-cleaved VEGF-C activity in the (A) Ba/F3-VEGFR-3/EpoR assay and (B) Ba/F3-VEGFR-2/EpoR assay. (C) Mature VEGF-D after secondary activation with cathepsin D in the Ba/F3-VEGFR-3/EpoR assay. (D) The minor form of mature VEGF-D generated by cathepsin D-cleavage is less active than the major mature form in the Ba/F3-VEGFR-2/EpoR assay. (E) KLK3 activation of VEGF-D increases its potency in the Ba/F3-VEGFR-2/EpoR assay. (F) The secondary activation of mature VEGF-C with cathepsin D led to a small decrease in the response of Ba/F3-VEGFR-2/EpoR cells (n = 2). Error bars indicate SD.

DOI: https://doi.org/10.7554/eLife.44478.022

The following source data and figure supplements are available for figure 7:

**Source data 1.** Ba/F3 assay showing the receptor-activating properties of cathepsin D-cleaved VEGF-C and VEGF-D.
DOI: https://doi.org/10.7554/eLife.44478.025

**Figure supplement 1.** Cathepsin D-cleaved pro-VEGF-D stimulates the phosphorylation of VEGFR-2 in PAE cells.
DOI: https://doi.org/10.7554/eLife.44478.023

**Figure supplement 2.** KLK3/PSA can proteolytically remove both propeptides from VEGF-D.
DOI: https://doi.org/10.7554/eLife.44478.024

complete (*Figure 6C*), whereas the cleavage of both pro-VEGF-C and mature VEGF-C was slower and incomplete even after 16 hr (*Figure 6AB*). As expected, the cathepsin D processing of mature VEGF-D abolished most of its activity in the Ba/F3-VEGFR-3/EpoR assay (*Figure 7C*) and reduced, but did not abolish its activity in the Ba/F3-VEGFR-2/EpoR assay (*Figure 7D*), while processing of pro-VEGF-D stimulated the phosphorylation of VEGFR-2 (*Figure 7—figure supplement 1*). KLK3 also activated pro-VEGF-D (*Figure 6D* and *Figure 7E*). When VEGF-D was produced from a full-length cDNA using the baculovirus system, a significant fraction of the protein did not undergo processing by proprotein convertases. This allowed us to observe two additional KLK3 cleavage sites in pro-VEGF-D. One of these cleavages has been reported previously (*Stacker et al., 1999a*); the other cleavage mimics the C-terminal cleavage catalyzed by proprotein convertases that cleave between the VHD and the N-terminal propeptide (*Figure 6D* and *Figure 7—figure supplement 2*).

## In vivo effects of the novel mature VEGF-C forms

To confirm that the two new forms of VEGF-C have also an effect *in-vivo*, we transduced mouse skeletal muscle (tibialis anterior) with recombinant adeno-associated viruses serotype 9 encoding the KLK3- or the cathepsin D- (CATD) form of VEGF-C. Both vectors stimulated lymphangiogenesis and angiogenesis (*Figure 8*). As expected on the basis of the binding and receptor phosphorylation results, the response to the KLK3-form was stronger compared to the cathepsin D-form. Both forms appeared to give a stronger response compared to the positive control (the ADAMTS3-form of VEGF-C), but the higher expression level of the shorter VEGF-C forms likely explains most of this difference (*Figure 8—figure supplement 1*).

# Discussion

Kallikrein-related peptidase 3 (KLK3) or prostate-specific antigen (PSA) is widely known as a prostate cancer marker (*Lilja et al., 2008*), which may also participate in prostate cancer development (*Koistinen and Stenman, 2012*). PSA/KLK3 is also the major protease responsible for seminal clot liquefaction, and thus plays a role in reproduction. In our study, we have found an unexpected link between these apparently separate functions in the form of Vascular Endothelial Growth Factor-C (VEGF-C) and VEGF-D.

## Requirement for VEGFs during reproduction

The angiogenic effect of VEGF-A is required for example for implantation (*Torry et al., 2007*) and corpus luteum formation (*Reynolds et al., 2000*). VEGF-A levels in human seminal plasma are variable, typically between 10–20 ng/ml (*Brown et al., 1995*; *Obermair et al., 1999*), and VEGF-A has been implicated as a fertility factor that acts on sperm cells (*Obermair et al., 1999*). Sperm motility has been reported to increase slightly as a response to VEGF-A (*Iyibozkurt et al., 2009*), and overexpression of a testis-specific VEGF-A transgene resulted in infertility (*Korpelainen et al., 1998*). VEGF-C is the lymphangiogenic counterpart of VEGF-A, and lymphangiogenesis is required for ovarian follicle maturation (*Rutkowski et al., 2013*), corpus luteum formation (*Abe et al., 2014*; *Nitta et al., 2011*) and uterine implantation (*Red-Horse, 2008*). Furthermore, VEGF-C and VEGF-D are hormonally regulated in the reproductive system (*Nitta et al., 2011*).

## KLK3/PSA as a VEGF-C activator

The prostate produces KLK3 and contributes active KLK3 to semen. KLK3 is the major protease in semen and participates in seminal clot liquefaction. KLK3 from human seminal plasma cleaved VEGF-C between its N-terminal propeptide and the VEGF homology domain. Compared to the major form of mature VEGF-C, the form produced by KLK3 lacks three amino acid residues from the N-terminus, but it still activated both VEGFR-2 and VEGFR-3. In vitro, both sperm liquefaction and VEGF-C exposure to KLK3 resulted in efficient cleavage of VEGF-C. However, in natural insemination, several factors, such as the vaginal environment or the absent mixing of the early prostatic fraction with the seminal vesicular fluid fraction (*Björndahl and Kvist, 2003*) may interfere with VEGF-C activation.

Apart from KLK3, seminal plasma also contains many other proteases involved in the proteolytic liquefaction cascade (*Emami and Diamandis, 2013*), which might contribute to VEGF-C activation (and inactivation), including cathepsin D (this study) and plasmin (*Jeltsch et al., 2014*; *Stief, 2007*).

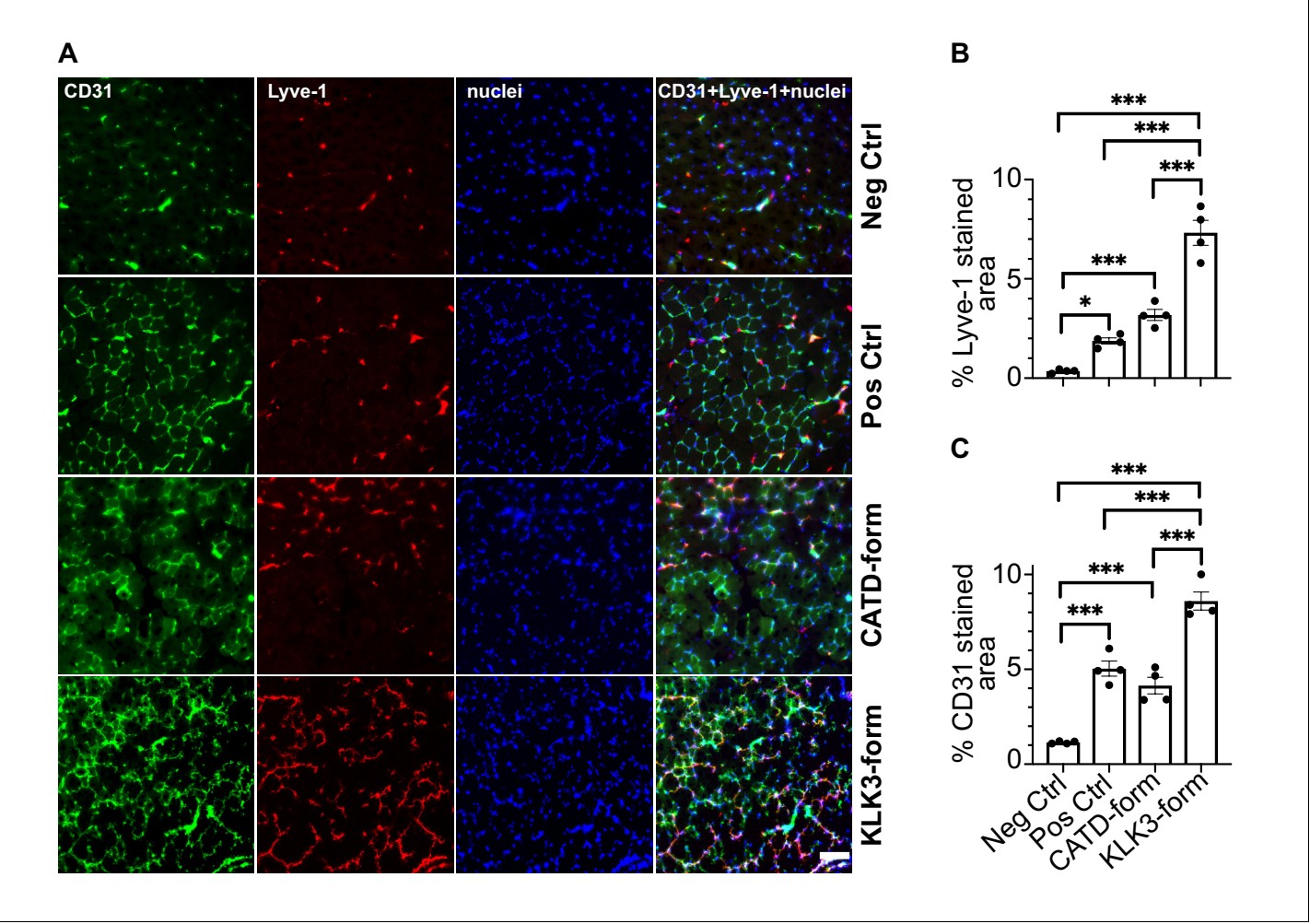

**Figure 8.** The KLK3- and cathepsin D-forms of VEGF-C induce lymphangiogenesis and angiogenesis in vivo. (**A**) Shown are the immunofluorescent stainings of the blood and lymphatic vessels in skeletal muscle transduced with recombinant adeno-associated virus subtype 9 (AAV9) encoding the KLK3- or the cathepsin D (CATD)-form of VEGF-C. Quantification of (**B**) Lyve-1 positive stained area and (**C**) CD31 positive stained area in AAV9 transduced tibialis anterior muscle. Data are presented as mean ± SEM, n = 4, one-way ANOVA with Tukey's *post hoc* test, \*\*\*p<0.001, \*p<0.05. Scale bar, 100 μm. (see also *Figure 8—figure supplement 1*).
DOI: https://doi.org/10.7554/eLife.44478.027

The following source data and figure supplement are available for figure 8:

**Source data 1.** In vivo quantification of lymphangiogenesis and angiogenesis induced by KLK3- and cathepsin D-forms of VEGF-C.
DOI: https://doi.org/10.7554/eLife.44478.029

**Figure supplement 1.** Expression of VEGF-C mRNA in tibialis anterior (TA) muscle.
DOI: https://doi.org/10.7554/eLife.44478.028

Similar to seminal plasma TGF-β (*Robertson et al., 2002*), which is also activated during liquefaction (*Emami and Diamandis, 2010*), VEGF-C might also contribute to the impregnation-associated immunomodulation. Several types of immune cells express VEGF-C receptors (*Hamrah et al., 2003*; *Krebs et al., 2012*; *Li et al., 2016*), and VEGF-C may be responsible for the immune tolerance of uterine NK cells during pregnancy (*Kalkunte et al., 2009*). However, since KLK3 exists only in higher primates (*Pavlopoulou et al., 2010*), any function of KLK3-mediated VEGF-C activation in seminal fluid is difficult to address experimentally. On the other hand, mice have many kallikrein-related peptidases that have no human counterparts (*Pavlopoulou et al., 2010*), and one of these might functionally replace KLK3 as an activator of VEGF-C. Unlike in mice, KLK3 prevents copulatory plug

formation in humans, where sperm liquefaction is thought to be a physical requirement for sperm movement (*Mann and Lutwak-Mann, 2012*).

## KLK3 and tumor (lymph)angiogenesis

VEGF-A inhibition marked a conceptual breakthrough in antiangiogenic cancer treatment (*Ferrara et al., 2005*). Although every single VEGF paralog in humans (PlGF, VEGF-B, VEGF-C, VEGF-D) has been proposed to mediate the tumor escape under anti-VEGF-A treatment (*Li et al., 2014*; *Lieu et al., 2013*), only VEGF-C and VEGF-D activate VEGFR-2 and VEGFR-3 (*Achen et al., 1998*; *Joukov et al., 1996*) and are therefore prime suspects (*Kubota, 2012*; *Li et al., 2014*; *Stacker et al., 2001*; *Wang and Tsai, 2015*). VEGF-A is able to activate VEGFR-2 immediately after secretion, but VEGF-C and VEGF-D need to be proteolytically processed to gain angiogenic (*Joukov et al., 1997*; *Stacker et al., 1999a*) or lymphangiogenic activity (*Jeltsch et al., 2014*).

The involvement of KLK3/PSA for tumor progression is still debated with studies arguing both in favor or against a tumor-promoting function of KLK3 (*Fortier et al., 1999*; *Ishii et al., 2004*; *LeBeau et al., 2010*; *Mattsson et al., 2008*; *Webber et al., 1995*). KLK3 expression is largely restricted to the male prostate (*MediSapiens Ltd, 2019*; *Shaw and Diamandis, 2007*), but small amounts can be found in other tissues, such as Skene's gland, the female homolog to the prostate (*Zaviacic and Ablin, 2000*). In pathological settings, the highest expression levels are found in prostate cancers (*MediSapiens Ltd, 2019*). We have hypothesized that KLK3 may facilitate early development of prostate cancer, but at later stages slow down cancer growth (*Koistinen and Stenman, 2012*). VEGF-C expression, which overlaps in the prostate with KLK3 expression (*Joory et al., 2006*), is similarly controversial, with some studies supporting (*Jennbacken et al., 2005*; *Yang et al., 2014*) and others refuting (*Mori et al., 2010*) its predictive ability for prostate cancer progression. Most experimental animal models confirm the role of VEGF-C for metastatic spread (*Brakenhielm et al., 2007*; *Burton et al., 2008*), and potential mechanisms have been identified in cell culture models (*Rinaldo et al., 2007*). This study shows that, at least in principle, KLK3 could contribute to the activation of tumor-derived VEGF-C or VEGF-D and thus to a (lymph)angiogenic tumor phenotype.

## How does CCBE1 accelerate VEGF-C activation?

KLK3 is a serine protease, but like ADAMTS3, its activity towards VEGF-C was increased by CCBE1. This reinforces the view that CCBE1 interacts with VEGF-C in the trimeric VEGF-C/ADAMTS3/CCBE1 complex, removing the masking of the cleavage site by the C-terminal domain of VEGF-C (*Joukov et al., 1997*). This idea is supported by the ability of the isolated C-terminal domain of VEGF-C to competitively inhibit CCBE1-accelerated VEGF-C activation by ADAMTS3 (*Jeltsch et al., 2014*; *Jha et al., 2017*). It would also explain why VEGF-C activation by plasmin is not controlled by CCBE1, as the plasmin cleavage site is located ~10 amino acids residues further away from the receptor binding epitopes than the ADAMTS3 and KLK3 cleavage sites (see *Figure 2*).

## Cathepsin D activates VEGF-C and VEGF-D with different outcomes

VEGF-C$_{DMH}$ was a designed variant with an N-terminal cleavage resembling that in the minor (VEGFR-2-specific) form of VEGF-D. After we had established that VEGF-C$_{DMH}$-like form is produced by cathepsin D via proteolytic cleavage of a longer VEGF-C polypeptide, we confirmed that also the VEGFR-2-specific form of VEGF-D (minor, mature form) is indeed produced by cathepsin D cleavage. However, cathepsin D cleavage affects VEGF-C and VEGF-D activities differently. While VEGF-D loses practically all binding affinity towards VEGFR-3, VEGF-C seems to lose preferentially its affinity towards VEGFR-2.

The minor mature form of VEGF-D was identified in the supernatant of VEGF-D-producing 293 cells (*Stacker et al., 1999a*), where VEGF-C$_{DMH}$ was not detected (*Joukov et al., 1997*), presumably because cathepsin D-cleavage of VEGF-C is inefficient. Alternatively, the ADAMTS3-cleavage of VEGF-C in 293 cells may have preemptively removed the recognition epitope required for cathepsin D cleavage.

## Secondary cleavage results in additional VEGF-C and VEGF-D species

Our data show that the longest forms of mature VEGF-C and VEGF-D can undergo secondary activation (i.e. N-terminally cleaved on top of a prior, activating cleavage). This introduces an additional layer of complexity into the regulation of VEGF-C and VEGF-D signaling since the cathepsin D-cleavage abolishes the VEGFR-3 binding of VEGF-D and reduces the VEGFR-2 binding of VEGF-C.

Cathepsin D is ubiquitously expressed, and although it is involved predominantly in lysosomal protein degradation (*Benes et al., 2008*), it can be secreted and soluble cathepsin D is found in saliva (our present findings) and in seminal plasma (*Jodar et al., 2016*). Secondary activation by cathepsin D may explain why we saw only weak VEGF-C-VEGFR-2 interaction when analyzing seminal plasma. It should be noted that cathepsin D has also been implicated in cancer metastasis (*Benes et al., 2008*; *Spyratos et al., 1989*), where VEGF-C can also play a role (*Karpanen et al., 2001*; *Mandriota et al., 2001*; *Skobe et al., 2001*). However, the cathepsin D-mediated secondary activation of the major, mature form of VEGF-D was very rapid, when compared to the very slow activation of VEGF-C (compare *Figure 6A and C*). Therefore, VEGF-D activation appears to be a more relevant function of cathepsin D than VEGF-C activation. The cathepsin D-processed minor form of mature VEGF-D showed a lower potency to activate VEGFR-2 than the major form of mature VEGF-D, likely reflecting the corresponding $K_D$ values (*Leppänen et al., 2011*). Despite this, as a net effect, cathepsin D cleavage of VEGF-D may result in increased angiogenic activity.

## The role of the N-terminal helix

In VEGF-A, the N-terminal helix in the VEGF homology domain appears essential for the receptor dimerization and activity (*Siemeister et al., 1998*), whereas the platelet derived growth factor does not need an N-terminal helix for receptor binding (*Muller et al., 1997*; *Shim et al., 2010*). The Leu119↓Lys120 (cathepsin D) cleavage of VEGF-C happens within the N-terminal helix, which contains binding epitopes for VEGFR-2 (*Leppänen et al., 2010*). The N-terminal helix also interacts with VEGFR-3. However, mutating the contacting amino acid residues Asp123 and Gln130 only ameliorates binding of VEGF-C to VEGFR-3 (*Leppänen et al., 2013*). The present receptor phosphorylation data strongly suggest that shortening of the helix leads to decreased activation of both VEGFR-2 and VEGFR-3, whereas a complete or near-complete removal of the N-terminal alpha helix- for example by extended plasmin exposure - abolishes all receptor binding. Inline with this, both the KLK3- and the cathepsin D-forms of VEGF-C induced lymphangiogenesis and angiogenesis in skeletal muscle. The N-terminal helix of VEGF-C is largely conserved among vertebrates, but the C-terminal end of the N-terminal propeptide and linker preceding the VEGF homology domain represent the most diverse sequences among VEGF-Cs in different species. These differences are especially noticeable between fish and the rest of the vertebrate clade (*Figure 2—figure supplement 2*), indicating potential differences in the VEGF-C activation.

## Separate activating proteases for each specific task?

Although ADAMTS3 appears to be responsible only for developmental lymphangiogenesis, our study indicates that other proteases may activate VEGF-C for specific niche functions, for example KLK3 in the reproductive system. The possible involvement of cathepsin D and KLK3 in tumor metastasis could be addressed in the appropriate gene-targeted mouse models. The possible other niche functions of VEGF-C, for example in the central nervous system (*Mackenzie and Ruhrberg, 2012*), in osmoregulation (*Machnik et al., 2009*) or in the immune system (*Loffredo et al., 2014*), may also be controlled by differentially regulated proteases.

# Materials and methods

**Key resources table**

| Reagent type (species) or resource | Designation | Source or reference | Identifiers | Additional information |
| --- | --- | --- | --- | --- |

*Continued on next page*

*Continued*

| Reagent type (species) or resource | Designation | Source or reference | Identifiers | Additional information |
|---|---|---|---|---|
| Cell line (*M. musculus*) | Ba/F3-hVEGFR-3/EpoR | *Achen et al., 2000* | | Murine pro-B cells expressing a chimeric VEGFR-3, from reference lab |
| Cell line (*M. musculus*) | Ba/F3-mVEGFR-2/EpoR | *Stacker et al., 1999b* | | Murine pro-B cells expressing a chimeric VEGFR-2, from reference lab |
| Cell line (*Sus scrofa domesticus*) | PAE-VEGFR-3-StrepIII | *Leppänen et al., 2013* | | Porcine aortic endothelial cells expressing strep-tagged VEGFR-3, from reference lab |
| Cell line (*Sus scrofa domesticus*) | PAE-VEGFR-2-StrepIII | *Anisimov et al., 2013* | | Porcine aortic endothelial cells expressing strep-tagged VEGFR-2, from reference lab |
| Cell line (*Sus scrofa domesticus*) | PAE-VEGFR-3 | *Pajusola et al., 1994* | | Porcine aortic endothelial cells expressing untagged VEGFR-3, from reference lab |
| Cell line (*Sus scrofa domesticus*) | PAE-VEGFR-2 | *Waltenberger et al., 1994* | | Porcine aortic endothelial cells expressing untagged VEGFR-2, from reference lab |
| Cell line (*Homo sapiens*) | 293T | ATCC | RRID:CVCL_0063 | Human embryonic kidney cells, from vendor |
| Cell line (*Cricetulus griseus*) | CHO DG44 | Invitrogen | | Chinese hamster ovary cells, from vendor |
| Cell line (*Drosophila melanogaster*) | Schneider S2 cells | Invitrogen | | Insect cells for protein production (*Drosophila* expression system), from vendor |
| Cell line (*Spodoptera frugiperda*) | Sf9 | Invitrogen | | Insect cells for protein production (FactBac system), from vendor |
| Transfected construct | pMT-Ex-VEGF-C-DMH | This paper. | 1271* | Production of the cathepsin D-cleaved form of VEGF-C in S2 cells |
| Transfected construct | pMT-hygro-BiPSP-hVEGF-C-FL | This paper. | 751* | Production of untagged pro-VEGF-C in S2 cells |
| Transfected construct | pSecTagI-IgKSP-ΔNΔC-hVEGF-C-H6 | This paper. | 2242* | Production of the plasmin1-cleaved form of VEGF-C (primary plasmin cleavage, between VEGF-C aa 102 and 103) |
| Transfected construct | pSecTagI-IgKSP-ΔNΔC-VEGF-C-ADAMTS3-H6 | This paper. | 2313* | Production of the ADAMTS3-cleaved form of VEGF-C (cleavage between aa residues 111 and 112) |
| Transfected construct | pSecTagI-IgKSP-ΔNΔC-VEGF-C-KLK3-H6 | This paper. | 2312* | Production of the KLK3-cleaved form of VEGF-C (cleavage between aa residues 114 and 115) |
| Transfected construct | pSecTagI-IgKSP-ΔNΔC-VEGF-C-CATD-H6 | This paper. | 2315* | Production of the cathepsin D-cleaved form of VEGF-C (cleavage between aa residues 119 and 120) |

*Continued*

| Reagent type (species) or resource | Designation | Source or reference | Identifiers | Additional information |
|---|---|---|---|---|
| Transfected construct | pSecTagI-IgKSP-ΔNΔC-VEGF-C-plasmin2-H6 | This paper. | 2314* | Production of the plasmin2-cleaved form of VEGF-C (secondary plasmin cleavage, between VEGF-C aa 127 and 128) |
| Transfected construct | pSecTagI-IgKSP-ΔNΔC-VEGF-C(C156S)-H6 | This paper. | 2318* | Production of the mature form of the VEGF-C-C156S mutant (primary plasmin cleavage, between VEGF-C aa 102 and 103) |
| Transfected construct | pMX-hCCBE1-StrIII | *Jeltsch et al., 2014* | 1494* | Production of full-length CCBE1 |
| Transfected construct | psubCAG-WPRE-IgKSP-ΔNΔC-hVEGF-C-KLK3 | This paper. | 2380* | Generation of recombinant adeno-associated virus |
| Transfected construct | psubCAG-WPRE-IgKSP-ΔNΔC-hVEGF-C-CATD | This paper. | 2351* | Generation of recombinant adeno-associated virus |
| Transfected construct | psubCMV-WPRE-IgKSP-ΔNΔC-hVEGF-C-ADAMTS3 | *Anisimov et al., 2009* | | Generation of recombinant adeno-associated virus (positive control) |
| Transfected construct | psubCMV-WPRE | *Paterna et al., 2000* | | Generation of recombinant adeno-associated virus (negative control) |
| Transformed construct | pFB1-melSP-hVEGF-D-FL-H6 | This paper and *Achen et al., 1998* | 229* | Generation of recombinant baculovirus (FastBac system) |
| Transformed construct | pFB1-melSP-ΔNΔC-hVEGF-D-H6 | This paper and *Achen et al., 1998* | 118* | Generation of recombinant baculovirus (FastBac system) |
| Biological sample (*H. sapiens*) | human saliva | collected from authors of this paper | | |
| Biological sample (*H. sapiens*) | human seminal plasma | collected from authors of this paper | | |
| Antibody | anti-VEGF-C antiserum, rabbit polyclonal | *Baluk et al., 2005* | AS no. 6 | WB (1:2000) |
| Antibody | anti-VEGF-C antiserum, rabbit polyclonal | *Joukov et al., 1997* | AS 882 | WB (1:1000) IP (1:500-1:1000) |
| Antibody | anti-VEGF-C antiserum, rabbit polyclonal | *Joukov et al., 1997* | AS 905 | WB (1:500) |
| Antibody | anti-VEGF-C antiserum, rabbit polyclonal | This paper. | AS 885 | WB (1:250) |
| Antibody | anti-VEGF-C antiserum, rabbit polyclonal | This paper. | AS 890 | WB (1:250) |
| Antibody | anti-VEGF-C antiserum, rabbit polyclonal | This paper. | AS no. 3/4 | WB (1:1000) |
| Antibody | anti-VEGF-C antibody, rabbit polyclonal | Abcam | RRID:AB_2241408 | WB (1:1000) |
| Antibody | anti-VEGF-C antibody, rabbit polyclonal | Abcam | ab135506 | WB (1:1000) |
| Antibody | anti-VEGF-C antibody, rabbit polyclonal | Novus/ Biotechne | NB110-61022 | WB (1:1000) |
| Antibody | anti-VEGF-C antibody, rabbit polyclonal | Cell Signaling Technology | RRID:AB_2213314 | WB (1:1000) |
| Antibody | anti-VEGF-C antibody, rabbit polyclonal | Invitrogen/ThermoFisher | RRID:AB_2547246 | WB (1:500) |

*Continued on next page*

*Continued*

| Reagent type (species) or resource | Designation | Source or reference | Identifiers | Additional information |
|---|---|---|---|---|
| Antibody | anti-VEGF-C antibody, rabbit polyclonal | Sigma-Aldrich/Merck | SAB1303101 | WB (1:500) |
| Antibody | anti-VEGF-C antibody, rabbit polyclonal | Sigma-Aldrich /Merck | SAB1303607 | WB (1:500) |
| Antibody | anti-VEGF-C antibody, goat polyclonal | R and D Systems /Biotechne | RRID:AB_2241406 | WB (1:1000) |
| Antibody | anti-VEGF-C antibody, mouse monoclonal | Santa Cruz Biotechnology | RRID:AB_11012156 | WB (1:500) |
| Antibody | anti-VEGF-C antibody, mouse monoclonal | Santa Cruz Biotechnology | RRID:AB_1131232 | WB (1:500) |
| Antibody | anti-VEGF-C antibody, mouse monoclonal | R and D Systems /Biotechne | RRID:AB_2213313 | WB (1:500) |
| Antibody | anti-VEGF-C antibody, mouse monoclonal | Invitrogen /ThermoFisher | RRID:AB_2725653 | WB (1:200) |
| Antibody | anti-VEGF-C antibody, mouse monoclonal | Sigma-Aldrich /Merck | SAB1306762 | WB (1:100) |
| Antibody | anti-KLK3 antibody, mouse monoclonal | *Stenman et al., 1999* | 5C7 | neutralization at 2-fold molar excess |
| Antibody | anti-VEGF-D, goat polyclonal | R and D Systems /Biotechne | RRID:AB_355293 | WB (1:1000) |
| Antibody | anti-phosphotyrosine antibody 4G10, mouse monoclonal | Millipore/ Merck | RRID:AB_309678 | WB (1:5000) |
| Antibody | anti-VEGFR-2, goat polyclonal | R and D Systems /Biotechne | RRID:AB_355320 | WB (1:1500) |
| Antibody | anti-VEGFR-3, mouse monoclonal | *Dumont et al., 1998* | 9D9F9 | WB (1:1000) |
| Antibody | anti-CCBE1, rabbit polyclonal | Atlas Antibodies /Sigma-Aldrich/ Merck | RRID:AB_10794515 | WB (1:1000) |
| Antibody | Penta·His Antibody, mouse monoclonal | Qiagen | RRID:AB_2619735 | WB (1:1500) |
| Antibody | anti-mouse Lyve-1, rabbit polyclonal | *Karkkainen et al., 2004* | | IF (1:1000) |
| Antibody | anti-mouse CD31, rat monoclonal | BD Biosciences | RRID:AB_393571 | IF (1:500) |
| Antibody | HRP-conjugated Strep-Tactin | IBA | 2-1502-001 | 1:100000 |
| Antibody | Donkey anti-goat IgG | Jackson Immuno Research | RRID:AB_2340390 | 1:2500 |
| Antibody | Goat anti-mouse IgG | Jackson Immuno Research | RRID:AB_10015289 | 1:2500 |
| Antibody | Goat anti-rabbit IgG | Jackson Immuno Research | RRID:AB_2313567 | 1:2500 |
| Antibody | Alexa 488 donkey anti-rat | Molecular Probes/ Thermo Fisher | RRID:AB_2535794 | 1:500 |
| Antibody | Alexa 594 donkey anti-rabbit | Molecular Probes/ Thermo Fisher | RRID:AB_141637 | 1:500 |
| Recombinant protein | KLK3 | *Wu et al., 2004*; *Zhang et al., 1995* | | isoform B |
| Recombinant protein | Cathepsin D (CATD) | R and D Systems/ Biotechne | 1014-AS | |

*Continued on next page*

*Continued*

| Reagent type (species) or resource | Designation | Source or reference | Identifiers | Additional information |
|---|---|---|---|---|
| Recombinant protein | pro-VEGF-C | This paper. | 751* | untagged pro-VEGF-C |
| Recombinant protein | ΔNΔC-VEGF-C or mature VEGF-C | *Kärpänen et al., 2006* | 792* | C-terminally histagged mature human VEGF-C (minor form) |
| Recombinant protein | VEGF-C$_{DMH}$ | This paper. | 2454* | DMH form of human VEGF-C expressed in S2 cells |
| Recombinant protein | pro-VEGF-D | *Achen et al., 1998* | 229* | C-terminally histagged human pro-VEGF-D |
| Recombinant protein | ΔNΔC-VEGF-D or mature VEGF-D | *Achen et al., 1998* | 118* | C-terminally histagged mature human VEGF-D (major form) |
| Recombinant protein | CCBE1-StrepIII | *Jeltsch et al., 2014* | 1494* | StrepIII-tagged human CCBE1 protein |
| Recombinant protein | VEGFR-3/Fc | *Jeltsch et al., 2006* | 810* | human VEGFR-3 extracellular domains 1–7 fused to IgG1Fc |
| Recombinant protein | VEGFR-2/Fc | *Leppänen et al., 2013*; *Leppänen et al., 2010* | 321* | human VEGFR-2 extracellular domains 1–3 fused to IgG1Fc |
| Commercial assay or kit | Human VEGF-C Quantikine ELISA Kit | R and D Systems/ Biotechne | DVEC00 | |
| Chemical compound, drug | streptactin resin/ sepharose | IBA | 2-1201-010 | |
| Chemical compound, drug | Protein A-Sepharose-4B beads CL-4B (PAS) | GE Healthcare | 17-0780-01 | |
| Chemical compound, drug | Chelex 100 | Bio-Rad | 1421253 | |
| Chemical compound, drug | cOmplete | Roche | 11697498001 | |
| Other | VECTASHIELD mounting medium with DAPI | Vector Laboratories | RRID:AB_2336790 | nucleic label |

*The asterisk denotes internal lab numbering of the corresponding DNA prep.

## Protein production and purification

KLK3 (isoform B) was purified by immunoaffinity chromatography from pooled seminal plasma (*Wu et al., 2004*). The separation of the different isoforms by anion-exchange chromatography was performed as described (*Zhang et al., 1995*). For the production of untagged pro-VEGF-C, full-length human VEGF-C cDNA was cloned into the Drosophila expression vector pMT-BiP (Invitrogen/Thermo Fisher Scientific, Waltham, MA). The protein was expressed in stably transfected S2 cells in Insect-Xpress medium (Lonza, Basel, Switzerland) supplemented with 250 µg/ml hygromycin at 26°C. The cells were induced with 1 mM CuSO$_4$ and the conditioned medium was harvested 4 days post-induction. VEGF-C was purified from the conditioned medium by Heparin affinity chromatography (HiTrap Heparin HP, GE Healthcare, Chicago, IL) at pH 6.7, followed by cation exchange chromatography over a MonoS or HiTrap SP HP column (GE Healthcare) at the same pH and gel filtration on a Superdex 200 Increase (GE Healthcare) column in PBS. C-terminally his-tagged pro-VEGF-D was produced with the baculovirus system as described (*Achen et al., 1998*). Purification was performed by affinity chromatography over Excel sepharose (GE Healthcare), followed by gel filtration as

described for pro-VEGF-C. C-terminally his-tagged mature VEGF-D was produced from a truncated cDNA analogous to pro-VEGF-D. CCBE1 protein was produced and purified as described (*Jeltsch et al., 2014*). Similarly, human VEGFR-3/Fc (containing extracellular domains 1–7) and VEGFR-2/Fc (containing extracellular domains 1–3) were purified as described (*Jeltsch et al., 2006*; *Leppänen et al., 2013*; *Leppänen et al., 2010*).

## Recombinant Adeno-Associated viral vector production

Recombinant Adeno-Associated Viruses (AAVs) were produced as previously described (*Jeltsch et al., 2014*).

## Cell lines

Stably transfected cell lines were obtained directly from the generating laboratories (indicated by the reference in the key resource table) and other cell lines were obtained from the indicated vendors, who authenticate and monitor for mycoplasma status of these products according to applicable regulations.

## Antibodies

All anti-VEGF-C antibodies are listed in the *Supplementary file 1*. We used further the following antibodies: anti-phosphotyrosine antibody 4G10 (Merck/Millipore), anti-VEGFR-3 antibody sc-321 (Santa Cruz Biotechnology, Dallas, TX), anti-VEGFR-2 (AF357, R and D Systems, Minneapolis, MN), anti-VEGF-D (AF286, R and D Systems), anti-CCBE1 (HPA041374, Atlas Antibodies/Sigma-Aldrich/ Merck), and Penta-His antibody (#34660, Qiagen, Hilden, Germany).

   For the immunofluorescence, the primary antibodies anti-CD31 (BD Biosciences) and anti-Lyve-1 (*Karkkainen et al., 2004*) were detected using the appropriate Alexa Fluor 488 and 594 secondary antibody conjugates (Molecular Probes/Invitrogen). Antiserum (AS) no. 3/4, AS 885 and AS 890 were generated like AS no. 6 (*Baluk et al., 2005*), except that mature VEGF-C (*Kärpänen et al., 2006*) was used as the antigen instead of pro-VEGF-C for AS no. 3/4, and peptide antigens (see *Supplementary file 1* for details) for AS 885 and AS 890.

## Activation of pro-VEGF-C and pro-VEGF-D by KLK3

0.94 µg of purified KLK3 was incubated with 1.7 µg of recombinant growth factor in TBS pH 7.7 at 37˚C for 24 hr, if not differently indicated. For blocking, the monoclonal antibody against KLK3, 5C7 (*Stenman et al., 1999*) was used in 2-fold molar excess and the cleavage was analyzed by SDS-PAGE/Western using antiserum 6 and 3/4 (VEGF-C) and AF286 (VEGF-D, R and D Systems). For CCBE1-enhanced cleavage experiments, 10 µl CCBE1-StrepIII (equal to the amount of CCBE1 purified from 12.5 ml of conditioned 293T medium) were included in the reaction.

## Activation of VEGF-C and VEGF-D by cathepsin D

80 µg of pro-VEGF-C/pro-VEGF-D in 240 µl PBS or ΔNΔC-VEGF-C/ΔNΔC-VEGF-D in 60 µl PBS were incubated with the same volume of human, recombinant cathepsin D, which had been activated and diluted according to the instructions of the manufacturer (1014-AS, R and D Systems). Incubation was performed at 37˚C, and aliquots were taken at 15 min, 1 hr, 4 hr and 16 hr and frozen at −80˚C until analysis. Samples were resolved by reducing SDS-PAGE and proteins were visualized by Coomassie Blue staining. The activation of pro-VEGF-D and ΔNΔC-VEGF-D was visualized by Western blotting.

## Transfections, Metabolic Labeling

293T and CHO cell transfections and procedures were performed as described (*Jeltsch et al., 2014*).

## Edman degradation

For the N-terminal sequence analysis, the digestion mixture of purified KLK3 and recombinant pro-VEGF-C or purified protein was resolved by SDS-PAGE and blotted to a PVDF membrane using 1xCAPS buffer/10% methanol. The membrane was Coomassie-stained and the band at 20 kDa was excised after destaining with 50% methanol. Edman degradation was performed using a Procise 494

HT sequencer (Applied Biosystems/Thermo Fisher Scientific) and data analyzed with the Sequence Pro software. Multiple N-termini were disambiguated by a fuzzpro search (*Rice et al., 2000*) of the major peaks against the VEGF-C and KLK3 sequences and eliminating results incompatible with the molecular weight observed on the gel.

## Ba/F3-VEGFR/EpoR Assays

The Ba/F3-hVEGFR-3/EpoR (*Achen et al., 2000*) and Ba/F3-mVEGFR-2/EpoR (*Stacker et al., 1999b*) bioassays were performed with recombinant proteins as described (*Mäkinen et al., 2001*).

## VEGF-C activation in seminal plasma

Fresh ejaculates, showing normal sperm parameters (*Cooper et al., 2010*), were collected from healthy volunteers among the authors (three different individuals) in full agreement with local regulations and institutional oversight. For analysis by SDS-PAGE/Western blotting, seminal plasma was separated from the cellular fraction and debris after approximately 30 min of liquefaction at RT by centrifuging twice for 10 min (at 1000 g and 10000 g). Seminal plasma was stored at −80°C until further analyses. Prior to analysis, thawed seminal plasma samples were sonicated and centrifuged again for 10 min at 16000 g at 4°C. The upper white layer was discarded and the clear fraction was collected for analyses.

To test the effects of divalent cation concentration and pH on the cleavage of VEGF-C in seminal plasma, 50 mg of Chelex 100 Resin (Bio-Rad, Hercules, CA), 10 µl 0.5M EDTA, or 25 µl 0.1M citric acid were added during the initial liquefaction to each ml of seminal fluid, and samples were incubated for 24 hr at 37°C before continuing with the centrifugation steps.

To slow the proteolytic liquefaction cascade, fresh ejaculates were immediately transferred to ice and a protease inhibitor cocktail (cOmplete, Roche) pre-dissolved in PBS was added at twice the recommended final concentration. Two centrifugation steps of 10000 g were performed at 4°C to separate the cellular and gel fraction from the liquid phase and samples were stored at −80°C until further analysis. Before the gel fraction was loaded, it was incubated at 37°C until liquefaction.

## Immunoprecipitation, SDS-PAGE, Western blotting and protein analysis

For precipitation with antibodies or soluble receptors, the seminal plasma samples (processed as described above) were diluted 1 + 1 with PBS and incubated with 30 µl protein A-Sepharose-4B beads and the respective antibody or soluble receptor overnight at 4°C. The beads were washed three times with PBS/0.05% Tween-20 and the bound proteins were eluted by adding 30 µl of 2X Laemmli standard buffer (LSB) followed by heating at 95°C for 10 min. For direct loading of proteins (digestion analysis of VEGF-C and VEGF-D, CCBE1 from seminal plasma), 2x or 5x LSB was added to the samples prior to boiling. For Western blotting, proteins were resolved on SDS-PAGE, transferred to PVDF membranes, blocked with 5% BSA in TBS-T for 1 hr and probed overnight with the relevant primary antibodies. The membranes were incubated with the appropriate HRP-conjugated secondary antibodies (Jackson Immuno Research, Cambridgeshire, UK, anti-rabbit IgG (111-035-003), anti-mouse IgG (115-035-003) or anti-goat IgG (705-035-003), 1:2500 in 5% skimmed milk in TBS-T) for 1 hr at RT and bands were visualized with ECL plus Western Blotting Substrate (Pierce/Thermo Fisher Scientific, Waltham, MA) or SuperSignal West Femto Maximum Sensitivity Substrate (Pierce/Thermo Fisher Scientific) using the LI-COR Odyssey Fc or cDigit Imaging System (Li_COR, Lincoln, NE). Direct visualization of proteins in the PAGE gels was performed by Coomassie Blue or silver staining.

## ELISA

The level of VEGF-C in seminal plasma (processed as described above) was estimated using the Human VEGF-C Quantikine ELISA Kit (DVEC00, R and D Systems) following the manufacturer's instructions.

## Stimulation of VEGFR-3 and VEGFR-2 phosphorylation

Near confluence, PAE cells expressing strep-tagged VEGFR-3 (*Leppänen et al., 2013*) or VEGFR-2 (*Anisimov et al., 2013*) were washed with PBS and starved for 4–5 hr in DMEM. PAE cells expressing untagged VEGFR-3 or VEGFR-2 starved for 16 hr in DMEM/0.1% BSA were used to analyze N-terminally truncated VEGF-Cs. The cells were stimulated for 10 min with sonicated centrifugation-cleared

seminal plasma diluted 1 + 1 with PBS (as described above), 20 ng/ml ΔNΔC-VEGF-C (*Kärpänen et al., 2006*) or equimolar amounts of N-terminally truncated VEGF-Cs (adjusted after quantification of VEGF-C levels in conditioned supernatant after transient transfection of CHO cells) to detect phosphorylation of VEGFR-3 and VEGFR-2. Then, the cells were washed twice with ice-cold PBS, lysed with modified RIPA buffer (50 mM Tris-HCl pH 8, 0.5% NP-40, 0.5% Triton X-100, EDTA-free protease inhibitor cocktail (cOmplete, Roche, Pleasanton, CA), 0.1 mM PMSF, 1 mM $Na_3VO_4$, and 1 mM NaF). VEGFR-3 and VEGFR-2 were precipitated from the cell lysate using Strep-Tactin Sepharose (IBA, Göttingen, Germany) for strep-tagged VEGFR-2/–3 or immunoprecipitated using protein A Sepharose (PAS) and anti-VEGFR-3 (clone 9D9F9, *Dumont et al., 1998*) or anti-VEGFR-2 (AF357, R and D Systems), washed three times with PBS/0.05% Tween-20/1 mM $Na_3VO_4$ and eluted with 2x Laemmli buffer and analyzed by SDS-PAGE/Western blot using the phospho-tyrosine-specific antibody 4G10 (Merck/Millipore, Darmstadt, Germany, 1:5000). Membranes were stripped using Re-Blot plus strong solution (Merck/Millipore) and re-probed with HRP-conjugated Strep-Tactin (IBA, 1:100000), anti-VEGFR-3 (9D9F9) or anti-VEGFR-2 (AF357) to verify equal loading.

## Fractionation of human saliva and VEGF-C cleavage activity assay

7 ml of filter-sterilized saliva collected from volunteers among the authors in agreement with local regulations were diluted 1 + 2 with running buffer (20 mM sodium acetate, pH 4.67) and loaded onto a MonoS column (GE Healthcare). After washing with running buffer, elution was performed with a linear 0–1M NaCl gradient and 1 ml fractions were collected. 20 µl of each fraction were diluted 1 + 4 with running buffer and 1.3 µg of pro-VEGF-C was added. After 36 hr incubation at 37°C, a Ba/F3-VEGFR-3/EpoR assay was performed with the samples.

## Interspecies analysis of VEGF-C sequences

Amino acid sequences of 40 VEGF-C orthologs representing all major vertebrate groups (fish, amphibians, reptiles, birds, mammals) were retrieved via a blastp search against human VEGF-C (UniProtKB P49767). To analyze clade-specific differences in the sequence context of the VEGF-C-activating cleavage, the sequences were truncated to include only sequences corresponding to human VEGF-C amino acids 55 to 228 (i.e. from the center of the N-terminal propeptide to the end of the VEGF homology domain). Alignment was performed with m_coffee (*Wallace et al., 2006*) and the sequences attached to the tip nodes of a phylogenetic species tree generated by opentree (*Hinchliff et al., 2015*). The results were rendered with the ETE toolkit (*Huerta-Cepas et al., 2016*). A Python script of the complete workflow is available from GitHub (*Jeltsch, 2018*; copy archived at https://github.com/elifesciences-publications/VEGFC).

## Mass spectrometric analysis

Six bands, with identical replicates, were cut from a Coomassie-stained SDS-PAGE gel. Samples were in-gel digested according to the standard protocols and analyzed by LC-ESI-MS/MS using the LTQ Orbitrap Velos Pro mass spectrometer (Thermo Fisher Scientific). The data files were searched for protein identification using Proteome Discoverer 1.4 software (Thermo Fisher Scientific) connected to a server running Mascot 2.4.1 (Matrix Science, Boston, MA). Data were searched against the SwissProt database (release 2014_01). The following search parameters were used: type of search - MS/MS Ion Search, taxonomy - human, enzyme - trypsin, fixed modifications - carbamidomethyl (C), variable modifications - oxidation (M), mass values - monoisotopic, peptide mass tolerance - ± 5 ppm, fragment mass tolerance - ± 0.5 Da, max missed cleavages - 1, instrument type - ESI-TRAP. Only proteins assigned at least with two unique peptides were accepted.

## Cloning

The pMX-hCCBE1-StrIII construct has been described before (*Jeltsch et al., 2014*). The S2 cell-expression vector pMT-Ex-VEGF-C-DMH was generated by deleting the 51 nucleotides coding for amino acids 103 to 119 of VEGF-C from pMT-Ex-ΔNΔC-VEGF-C-$H_6$, a modified pMT/BiP/V5-His C vector (Invitrogen/Thermo Fisher Scientific), expressing mature VEGF-C (*Kärpänen et al., 2006*). pMT-hygro-BiPSP-hVEGF-C-FL (for the production of untagged pro-VEGF-C) was generated by PCR-amplification of sequences corresponding to amino acids 32–419 of VEGF-C and cloning of the product into BglII-opened pMT-BiPV5HisC-hygro, another derivative of pMT/BiP/V5-His C, in which

the 260 bp SapI-AccI fragment had been replaced by the SapI-AccI hygromycin expression cassette from pCoHygro (Invitrogen/Thermo Fisher Scientific).

pSecTagI-IgKSP-$\Delta$N$\Delta$C-hVEGF-C-H$_6$ (the mammalian vector expressing mature VEGF-C corresponding to VEGF-C activated by plasmin cleavage between VEGF-C amino acids 102 and 103) was constructed by inserting the BamHI/BclI-cut VEGF-C PCR amplification product of primers 5'-GATGCTCGAGGATCCGACAGAAGAGACTATAAAATTTGC-3' and 5'-GCATGATCACAGTTTAGACATGC-3' into the BamHI-opened pMosaic vector (Jeltsch et al., 2006). The cDNAs coding for N-terminally truncated VEGF-C (corresponding to mature VEGF-C forms as activated by KLK3 cleavage, ADAMTS3 cleavage, and plasmin cleavage between amino acid residues 127 and 128) were PCR amplified from pSecTagI-IgKSP-$\Delta$N$\Delta$C-hVEGF-C-H$_6$ using specific forward primers (5'-TCCG GATCCGGATCCAAATACAGAGATCTTGAAAAGTATTGATAATGAGTGG-3'; 5'-TC CGGATCCGGATCCAGCACATTATAATACAGAGATCTTGAAAAGTATTG-3'; and 5'-TCCGGATCCGGATCCAAAGACTCAATGCATGCCACG-3') and the same reverse primer (5'-ACCTACTCAGACAATGCGATGC-3'), and subcloned into pSecTagI-IgKSP-$\Delta$N$\Delta$C-hVEGF-C-H$_6$ as BamHI-EcoRI fragments. The DMH/CatD form and C156S mutant of VEGF-C were subcloned in the same fashion from pMT-Ex-VEGF-C-DMH and pREP7-VEGF-C-C156S (Joukov et al., 1998) into the same vector (using forward primers 5'-CGGATCCAAAAAGTATTGATAATGAGTGGAGA-3' and 5'-GCGGATCCGACAGAAGAGACTATAAAA-3' and reverse primer 5'-GGAATTCAATGATGATGATGGTGATGCAGTTTAGACATGC-3').

The shuttle vectors to produce pro-VEGF-D (pFB1-melSP-hVEGF-D-FL-H$_6$) and a mature form of VEGF-D (pFB1-melSP-$\Delta$N$\Delta$C-hVEGF-D-H$_6$) with the baculovirus system were generated by restriction-cloning the BamHI/HindIII-fragments of the PCR products of primers 5'-TGCGGATCCCTCCAGTAATGAACATGGACCAGTGAAGCGATC-3' and 5'-GACAAGCTTAATGATGATGATGGTGATGAGGATTCTTTCGGCTGTGGGGC-3' (for pro-VEGF-D) and 5'-TGCGGATCCGTCAGCATCCCATCGGTCCACTAGGTTTG-3' and 5'-GACAAGCTTAATGATGATGATGGTGATGGGGGGCTGTTGGCAAGCACTTAC-3' (for mature VEGF-D) into a modified pFASTBAC1 vector (Gerhardt et al., 2003).

## Cloning of AAV9 constructs

The sequence coding for the Immunoglobulin Kappa signal peptide was amplified using forward primer 5'-CTAAAAGCTGCGGAATTGTACCCGCGGCCGCTAGCGCCACCATGGAGACAGAC-3' and reverse primer 5'-GTCACCAGTGGAACCTGG-3' and the VEGF-C CDS was amplified using forward primer 5'-CTGCTCTGGGTTCCAGGTTCCACTGGTGACAAAAGTATTGATAATGAGTGGAGAAAGAC-3' and reverse primer 5'-AAATTTTGTAATCCAGAGGTTGATTATCGACGCGTTCAACGTCTAATAATGGAATGAACT-3'. Both fragments were assembled into a MluI- and NheI-opened and CIPped psubCAG-WPRE vector (Weltner et al., 2012) resulting in psubCAG-WPRE-IgKSP-$\Delta$N$\Delta$C-hVEGF-C-CATD. psubCAG-WPRE-IgKSP-$\Delta$N$\Delta$C-hVEGF-C-KLK3 was assembled as above, but the reverse primer for the Immunoglobulin Kappa signal peptide CDS amplification was replaced by 5'-TTATCAATACTTTTCAAGATCTCTGTATTGTCACCAGTGGAACCTGG-3' and the forward primer for VEGF-C CDS amplification by 5'-CAATACAGAGATCTTGAAAAGTATTGATAATG-3'.

## In vivo experiments

AAV9s (dose of $4 \times 10^{10}$ in 40 µl) encoding negative control, positive control (ADAMTS3-cleaved form of VEGF-C), KLK3-cleaved form of VEGF-C (KLK3-form) and Cathepsin D-cleaved form of VEGF-C (CATD-form) were injected into the Tibialis anterior (TA) muscles of C57Bl/6JRccHsd (Envigo Harlan) female mice. Mice were sacrificed 3 weeks after transduction and the tibialis muscles were harvested. All animal experiments carried out in this study were performed according to guidelines and regulations approved by the National Board for Animal Experiments of the Provincial State Office of Southern Finland.

## Histochemistry and immunofluorescence

Mouse tibialis anterior muscle samples were embedded into Tissue-Tek OCT and frozen in liquid nitrogen-cooled isopentane. 10µm-sections were stained for the lymphatic marker Lyve-1 (Karkkainen et al., 2004, 1:1000) and blood vascular marker CD31 (BD Biosciences, San Jose, CA, 1:500), followed by Alexa-conjugated secondary antibodies (Molecular Probes/Thermo Fisher Scientific). Nuclei were stained with DAPI with VECTASHIELD (Vector Laboratories, Burlingame, CA).

Fluorescent images were obtained with an Axio Imager Z2 upright epifluorescence microscope (Carl Zeiss AG, Oberkochen, Germany). Images were processed and analysed with Fiji ImageJ (NIH).

## RNA extraction and quantitative real time PCR

Muscle tissues were lysed using Trisure reagent (Bioline, London, UK) and the RNA was extracted with Nucleospin RNA II kit (Macherey-Nagel, Düren, Germany). cDNA was synthesized with High-Capacity cDNA Reverse Transcription Kits (Applied Biosystems/Thermo Fisher Scientific) using 1 µg RNA. qRT-PCR was performed with SensiFast SYBR No-ROX Kit (Bioline). All data were normalized to GAPDH. Relative gene expression levels were calculated using the $2^{-\Delta\Delta Ct}$ method. VEGF-C (fwd 5'-TGAACACCAGCACGAGCTAC-3', rev 5'-TCGGCAGGAAGTGTGATTGG-3') and mGAPDH (fwd 5'-ACAACTTTGGCATTGTGGAA-3', rev 5'-GATGCAGGGATGATGTTCTG-3') primers were used for the real time PCR.

## Statistical analysis

Data are presented as mean ± SD or mean ± SEM. Data were analysed using GraphPad Prism statistical analysis software (Version 8). Data analysis details are mentioned in the respective figure legends.

## Acknowledgements

We thank Tapio Tainola for DNA sequencing, Maria Arrano de Kivikko for technical help with the animal experiments, Sini Miettinen from the Proteomics Unit of the Institute of Biotechnology (University of Helsinki, Finland) for protein sequencing, and Anne Rokka from the Turku Proteomics Facility (Turku Centre for Biotechnology, Finland) for the mass spectrometric analysis. We further thank Seppo Kaijalainen for the cloning of the AAV constructs, the Biomedicum Imaging Unit, Tanja Laakkonen from the AAV Gene Transfer and Cell Therapy Core Facility of Biocenter Finland, and the Laboratory Animal Centre of the University of Helsinki for professional services.

## Additional information

### Funding

| Funder | Grant reference number | Author |
|---|---|---|
| Academy of Finland | 265982 | Michael Jeltsch |
| Finnish Foundation for Cardiovascular Research | | Michael Jeltsch |
| Jane ja Aatos Erkon Säätiö | | Michael Jeltsch |
| Cancer Society of Finland | | Sawan Kumar Jha Michael Jeltsch |
| Magnus Ehrnroothin Säätiö | | Michael Jeltsch |
| K. Albin Johanssons Stiftelse | | Michael Jeltsch |
| University of Helsinki | Integrated Life Science Doctoral Program | Sawan Kumar Jha |
| Sigrid Jusélius Foundation | | Hannu Koistinen |
| Laboratoriolääketieteen Edistämissäätiö | | Hannu Koistinen |
| European Research Council | Horizon 2020 Research and Innovation programme 743155 | Kari Alitalo |
| Wihuri Research Institute | | Sawan Kumar Jha Kari Alitalo |
| Academy of Finland | Centre of Excellence Program 2014-2019, 307366 | Kari Alitalo |
| Novo Nordisk Foundation | | Kari Alitalo |

| Biomedicum Helsinki-säätiö | | Sawan Kumar Jha |
| Päivikki and Sakari Sohlberg Foundation | | Khushbu Rauniyar |
| Academy of Finland | 272683 | Michael Jeltsch |
| Academy of Finland | 273612 | Michael Jeltsch |
| Academy of Finland | 273817 | Michael Jeltsch |

The funders had no role in study design, data collection and interpretation, or the decision to submit the work for publication.

### Author contributions
Sawan Kumar Jha, Conceptualization, Formal analysis, Funding acquisition, Investigation, Visualization, Methodology, Writing—original draft, Writing—review and editing; Khushbu Rauniyar, Formal analysis, Supervision, Funding acquisition, Validation, Investigation, Visualization, Methodology, Writing—original draft, Writing—review and editing; Ewa Chronowska, Investigation, Visualization, Writing—review and editing; Kenny Mattonet, Conceptualization, Funding acquisition, Investigation, Visualization, Writing—review and editing; Eunice Wairimu Maina, Investigation, Visualization, Methodology; Hannu Koistinen, Ulf-Håkan Stenman, Conceptualization, Resources, Writing—review and editing; Kari Alitalo, Resources, Supervision, Funding acquisition, Writing—review and editing; Michael Jeltsch, Conceptualization, Software, Supervision, Funding acquisition, Investigation, Methodology, Writing—original draft, Project administration, Writing—review and editing

### Author ORCIDs
Sawan Kumar Jha https://orcid.org/0000-0003-1898-4928
Khushbu Rauniyar https://orcid.org/0000-0001-5485-7040
Kenny Mattonet https://orcid.org/0000-0002-9705-8086
Michael Jeltsch https://orcid.org/0000-0003-2890-7790

### Ethics
Animal experimentation: All animal experiments carried out in this study were performed according to guidelines and regulations approved by the National Board for Animal Experiments of the Provincial State Office of Southern Finland (ESAVI/7012/04.10.07/2016).

### Decision letter and Author response
Decision letter https://doi.org/10.7554/eLife.44478.033
Author response https://doi.org/10.7554/eLife.44478.034

## Additional files

### Supplementary files
• Supplementary file 1. List of anti-VEGF-C antibodies used in this study. The data for the antibodies in this list were obtained from the product description provided by the supplier. Data missing from suppliers' websites were obtained by direct request to customer support. The dilution refers to what was used for the Western blot analyses performed in this study and is identical to the supplier's recommendation for commercially available antibodies.
DOI: https://doi.org/10.7554/eLife.44478.030

• Supplementary file 2. Results of mass spectrometric analysis of enriched VEGF-C cleaving activity. Protein profile of the six samples excised from the SDS-PAGE gel as obtained by LC-ESI-MS/MS analysis.
DOI: https://doi.org/10.7554/eLife.44478.026

• Transparent reporting form
DOI: https://doi.org/10.7554/eLife.44478.031

## Data availability

All data generated or analysed during this study are included in the manuscript and supporting files. Python scripts are available from https://github.com/mjeltsch/VEGFC (copy archived at https://github.com/elifesciences-publications/VEGFC).

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
