## [Decision Letter]

Thank you for submitting your article "KLK3/PSA and Cathepsin D activate VEGF-C and VEGF-D" for consideration by *eLife*. Your article has been reviewed by three peer reviewers, Gou Young Koh as the Reviewing Editor and Reviewer #1, the evaluation has been overseen by a Reviewing Editor and Didier Stainier as the Senior Editor.

The reviewers have discussed the reviews with one another and the Reviewing Editor has drafted this decision to help you prepare a revised submission

The comments are relatively favorable and constructive. However, the reviewers are concerned by the low numbers of sample sizes and lack of statistical analyses. They also request a more biological context on the importance of KLK3 cleavage, which should be definitely and additionally included in the revised manuscript. Moreover, the approval and clear statement on institutional oversight of the process of sperm donation within the lab are required. I believe the authors could readily address most of the comments, but please provide the reasons for not implementing the suggested changes where necessary.

*Reviewer #1:*

The study by Sawan Kumar Jha and Khushbu Rauniyar et al. provides mechanistic and molecular insights into the activation of VEGF-C and VEGF-D. The authors demonstrated that the KLK3/PSA and cathepsin D, the proteases in human body fluids, can activate VEGF-C and VEGF-D. Moreover, they found that cathepsin D is able to further activate mature forms of VEGFC and VEGF-D, which alters their binding preferences to VEGFR-2 or VEGFR-3. These findings are novel and intriguing and the data are well presented. Importantly, the study provides a novel potential regulatory mechanism underlying angiogenesis and lymphangiogenesis during development or tumor progression. Therefore, I consider this work suitable for publication in eLife after addressing the following minor points.

1) In Figure 3, the authors measured VEGF-C content in human seminal plasma and demonstrated that pro-VEGF-C become a mature VEGF-C by protease activity, supposedly of KLK3. Because KLK3/PSA is the major protease in human seminal fluid, these results suggest that KLK3-mediated production of a novel VEGF-C might have biological significances during human reproduction and progression of prostate cancer. However, the authors did not show VEGF-D content and its activation in the seminal fluid, although they demonstrated in Figure 6 that cathepsin D, which is also present in human seminal plasma, can activate VEGF-D more efficiently than VEGF-C in vitro. I am sure the authors wish to examine further on the VEGF-D content and activation in the human seminal fluid.

2) Figure 3C, lane 4: additional explanation of the experimental condition (37℃, 5mM EDTA) and results (in comparison with lane 2 (Chelex-100)) should be included in the manuscript or figure legend.

3) In Figure 4, the authors demonstrate that VEGF-C activation by KLK3 is enhanced by CCBE1. However, the presence or absence of CCBE1 in each lane is not indicated.

4) Figure 3B legend "Seminal VEGF-C does stimuspand late the phosphorylation of VEGFR-2..": The word 'stimuspand' needs to be corrected.

*Reviewer #2:*

This manuscript carefully analyses molecular mechanisms driving the proteolytic processing and activation of VEGF-C in the reproductive system, and the key role of KLK3 in these processes. In general the data are novel and of good quality, and the claims are well supported by the data. However, the biological significance of the findings is somewhat limited by our lack of insight into the functional importance of seminal VEGF-C in the reproductive system – the authors speculate a potential role in impregnation-associated immunomodulation. The following issues need to be addressed:

1) I could not interpret Figure 4. In which samples was CCBE1 included and in which was it not included?

2) The legend to Figure 1 does not indicate the expected sizes of pro-VEGF-C and mature VEGF-C, nor what the Ba/F assays are assessing.

3) To my knowledge the Ba/F assays allow indirect monitoring of the binding and cross-linking of the VEGF receptor extracellular domains, which is useful information. But the Authors should not state in the text that this demonstrates receptor activation which should be assessed by monitoring receptor phosphorylation.

4) The legend to Figure 7 must state what type of assays were used to generate these data.

5) The Discussion deals with too many themes and is way too long – it needs a major restructure to ensure it focuses predominantly on the key issues.

*Reviewer #3:*

Kumar Jha and colleagues describe a VEGF-C isoform found in seminal fluid and go on to identify the protease KLK3 as responsible for the processing of this protein in this bodily fluid. The authors go on to show, that like in other tissues for other VEGF-C cleavage events, the cleavage of VEGF-C by KLK3 is enhanced by CCBE1. Comparison of the binding capacities of the different N-terminal cleavage products of VEGF-C reveal different potentials for different forms. They also identified Cathepsin D in human saliva as an additional protease that can activate VEGF-C and downstream signalling, further increasing the different mechanism with potential to activate and modify VEGF-C. The authors also contrast KLK3 an Cathepsin functions at the level of VEGF-D activation.

The activation of VEGF-C and VEGF-D by proteases is a crucial step in VEGF-R signalling in normal physiology, development and in pathological settings. These findings will be of interest to researchers in angiogenesis and lymphangiogenesis as they identify only the third protease (KLK3) known to cleave VEGF-C. However, the scope of the paper is somewhat limited. While these findings appear to be novel, the study only investigates biochemical capability of KLK3 and the biological relevance of the role of KLK3 in VEGF-C cleavage is not shown. It is unclear if this cleavage and role of KLK3 is important in sperm formation, maturation or function or if it may even have expression and function more broadly. There are also many data points lacking quantification, reliable numbers of repeats and statistical support. To more convincingly support the manuscripts conclusions, the authors should address the concerns/comments indicated below.

Major comments:

1) While the data appears convincing in the representative cases shown, many of the claims made in the manuscript are not backed by large datasets, lack sufficient numbers of repeats, quantification of western blot data and statistics.

Some examples:

1.1) In Figure 3 (N=2 represented with western blot data) and the supporting text, the authors state "Chelation of KLK3-inhibitory Zn2+ ions with Chelex 100 increased the yield of mature VEGF-C as did lowering the pH with citric acid", however it’s not clear from the data in Figure 3C that this is the case. This data needs to be quantified and expressed with error bars as graphs with statistical analysis. The data should be displayed as the ratio of mature-VEGF-C to pro-VEGF-C.

1.2) Figure 5B is also n=2 westerns with no quantification/densitometry analysis and statistical support (p-values). Further quantification and potentially repeats are needed.

1.3) Figure 6 (n=2) lacks quantification/densitometry and p-values. Further quantification and potentially repeats are needed.

1.4) In Figure 4, it is unclear with the current labelling which lanes contain CCBE1. Furthermore if the labels are correct, the experiment lacks a KLK3 only control and a negative control (no KLK3 and no CCBE1). Without the full panel of controls and careful quantification (densitometry analysis and statistical support (p-values)) it is not possible to know if processing is additive or enhanced and the claims are not fully supported by the data.

2) The study provides exclusively evidence for a biochemical capability of KLK3 in VEGF-C regulation and thus is limited in scope. The biological, cellular and tissue level outcomes, consequences and phenotypes associated with the cleavage of VEGFC by KLK3 are not presented in vitro or in vivo. Further information would substantially increase the interest in the study. Does the absence of KLK3 cleavage impact sperm formation or function? Or exogenous administration of KLK3 cleaved vs other forms of VEGF-C differentially impact sperm biology? Is there any evidence that the loss of VEGFC in fact impacts on the formation, maturation or function of sperm?

3) The authors discuss the possibility that KLK3 could be relevant in a tumour setting, which would greatly enhance the scope and interest in the study, however they fail to investigate if this is likely. The manuscript would greatly benefit from an analysis of the distribution of KLK3 in other tissues, especially in pathological tissues and tumours. Analysis of KLK3 distribution in pathological and normal tissues relative to VEGFC distribution would greatly expand the relevance of these findings and interest to researchers.

4) Given that KLK3 can activate VEGF-D, is VEGF-D present in seminal fluid and is the cleavage of VEGF-D by KLK3 enhanced by CCBE1?

5) It would be interesting in the context of the paper to know if, similar to cleavage by KLK3, whether cleavage of VEGF-C and VEGF-D by Cathepsin D is enhanced by CCBE1?

6) The authors show that the different N-terminal cleaved forms of VEGFC have altered binding to VEGFR2 or VEGFR3. This is done at the level of pull-down with Fc forms of R2 and R3. Does this correlate with altered signalling at these receptors for the different ligand forms? Can the authors compare the bioactivity of the different cleavage products shown in Figure 2 at the level of phosphorylation or cell signalling/biology?

7) In Figure 6, the authors state that the cleavage of ΔNΔC-VEGF-C by Cathepsin D leads to "super-activation". They show that the processed form is cleaved into even smaller forms in the presence of Cathepsin D. Although Catherpsin D enhances signalling in Figure 7, the claim further processing of ΔNΔC-VEGF-C is hyperactivation at the level of the ligand seems misleading. It is not backed up by any biochemical data on phosphorylation of VEGFR-3 driven by the specific smaller forms of VEGF-C. Could this additional cleavage not be rendering an active form less active? Can the smaller forms be tested in isolation?

---

## [Author Response]

Reviewer #1:1) In Figure 3, the authors measured VEGF-C content in human seminal plasma and demonstrated that pro-VEGF-C become a mature VEGF-C by protease activity, supposedly of KLK3. Because KLK3/PSA is the major protease in human seminal fluid, these results suggest that KLK3-mediated production of a novel VEGF-C might have biological significances during human reproduction and progression of prostate cancer. However, the authors did not show VEGF-D content and its activation in the seminal fluid, although they demonstrated in Figure 6 that cathepsin D, which is also present in human seminal plasma, can activate VEGF-D more efficiently than VEGF-C in vitro. I am sure the authors wish to examine further on the VEGF-D content and activation in the human seminal fluid.

We performed pull-down of VEGF-D from seminal fluid using VEGFR-2/Fc and straight loading, but we did not detect VEGF-D in seminal fluid within the detection limit of the anti-VEGF-D antibody (R&D Systems AF286). We mention now in the manuscript in the Results section "Human seminal fluid contains VEGF-C", that we did not detect VEGF-D in seminal plasma and give the corresponding data in supplemental Figure 5. However, we only used one antibody and we cannot exclude that VEGF-D is present in low amounts as we are limited to the sensitivity of the antibody that we used.

2) Figure 3C, lane 4: additional explanation of the experimental condition (37℃, 5mM EDTA) and results (in comparison with lane 2 (Chelex-100)) should be included in the manuscript or figure legend.

We have added the information to the figure legend. Additionally, we have quantified the ratios between pro-VEGF-C and activated VEGF-C in order to estimate the reliability of these results. We indeed found that the change upon ion sequestering is not statistically significant and added this information to the text.

3) In Figure 4, the authors demonstrate that VEGF-C activation by KLK3 is enhanced by CCBE1. However, the presence or absence of CCBE1 in each lane is not indicated.

A cropped version of the figure has somehow slipped into the final manuscript... Now everything should be ok!

4) Figure 3B legend "Seminal VEGF-C does stimuspand late the phosphorylation of VEGFR-2..": The word 'stimuspand' needs to be corrected.

Corrected.

Reviewer #2:[…] The following issues need to be addressed:1) I could not interpret Figure 4. In which samples was CCBE1 included and in which was it not included?

Resolved (same issue as raised by reviewer 1, item 3).

2) The legend to Figure 1 does not indicate the expected sizes of pro-VEGF-C and mature VEGF-C, nor what the Ba/F assays are assessing.

We have added the expected sizes of the cleavage products based on previous literature (Joukov et al. 1997). We also indicate now the different forms of VEGF-C by arrows/asterisk. We also indicate now what the Ba/F3 assay is measuring: "VEGF-C processed by KLK3 is biologically active as shown by Ba/F3 assays, which translate activation of a hybrid VEGFR/EpoR receptor into cell survival".

3) To my knowledge the Ba/F assays allow indirect monitoring of the binding and cross-linking of the VEGF receptor extracellular domains, which is useful information. But the Authors should not state in the text that this demonstrates receptor activation which should be assessed by monitoring receptor phosphorylation.

The reviewer is correct that the Ba/F3 assay does not always faithfully replicate the actual receptor phosphorylation potential. Therefore, we have replaced all occurrences where we equate Ba/F3 results with receptor activation. The first occurrence is: "and found that it is able to activate both VEGFR-2 and VEGFR-3" replaced by "and found that it promoted the survival of both VEGFR-2/EpoR and VEGFR-3/EpoR cells". We further changed the figure labelling in Figures 1 and 7 from VEGFR to VEGFR/EpoR to indicate that a hybrid receptor was used to test the receptor activation potential. Similarly, we changed the wording in the subsection “Cathepsin D activates both VEGF-C and VEGF-D” and in the legend of Figure 7 to clarify, that we used chimeric VEGFR/EpoR receptors as proxies for the native receptors.

The known cases where Ba/F3 assay and actual receptor phosphorylation disagree have been pinpointed to exceptional geometric arrangements of the two receptor binding interfaces of the ligands (e.g. high affinity binders of EpoR, which do not at all or only partially activate the receptor). Since all VEGFs are dimers, and since the congruence of Ba/F3 assay data with receptor phosphorylation data has been shown repeatedly, it is reasonable to assume, that the Ba/F3 assay data shown here largely reflect receptor phosphorylation data. However, in the absence of crystal structure data for the newly

described VEGF-C species, we have to agree with the reviewer, since these VEGF-C or VEGF-D species might display distortions compared to the known structures.

We have also added experimental data using stimulation of phosphorylation of the native receptors for all different mature forms of VEGF-C and these results support the notion that, at least for these molecules, the Ba/F3 assays mirror the receptor phosphorylation.

4) The legend to Figure 7 must state what type of assays were used to generate these data.

We have inserted a sentence at the beginning of the figure legend to indicate the type of assay.

5) The Discussion deals with too many themes and is way too long – it needs a major restructure to ensure it focuses predominantly on the key issues.

We have restructured and shortened the Discussion from ~3000 words down to less than 1800 words, reducing its length by more than four pages. We made the following major changes:

1) We have moved the entire paragraph "Failure to detect seminal VEGFs by protein mass spectrometry" to the supplementary data (Figure 3—figure supplement 3 legend).

2) We have shortened the paragraph "KLK3 and tumor (lymph)angiogenesis" while integrating the additional data about KLK3 expression, that was specifically requested by reviewer

3) We have removed the first part of the paragraph that discusses the anti-VEGF-C antibody sensitivity ("Detecting active versus inactive VEGF-C").

4) We have moved the second part of the same paragraph into the legend for the supplementary figure, that shows the sensitivity testing results.

5) We merged and shortened the paragraphs "VEGF-C DMH preferentially binds VEGFR-3" and "Cathepsin D activates VEGF-C and VEGF-D" into a new paragraph "Cathepsin D activates VEGF-C and VEGF-D with different results".

6) We have merged and shortened the paragraphs "KLK3/PSA as a VEGF-C activator" and "Differences in the kallikrein protease family between humans and mice".

7) We have removed the speculation that cathepsin D might be a better VEGFR-3-specific activator than the engineered VEGFR-3-specific form of VEGF-C ("VEGF-CC156S).

8) We have shortened the paragraph about the mode of CCBE1 action by removing the arguments against the "unmasking hypothesis".

Reviewer #3:Major comments.1) While the data appears convincing in the representative cases shown, many of the claims made in the manuscript are not backed by large datasets, lack sufficient numbers of repeats, quantification of western blot data and statistics.Some examples:1.1) In Figure 3 (N=2 represented with western blot data) and the supporting text, the authors state "Chelation of KLK3-inhibitory Zn2+ ions with Chelex 100 increased the yield of mature VEGF-C as did lowering the pH with citric acid ", however it’s not clear from the data in Figure 3C that this is the case. This data needs to be quantified and expressed with error bars as graphs with statistical analysis. The data should be displayed as the ratio of mature-VEGF-C to pro-VEGF-C.

This was clearly us not paying attention, thanks for catching this! The repeated experiments showed indeed that the enhanced activation by CHELEX/EDTA are not statistically significant as well as the pH shift. We have added the quantification to Figure 3 and modified the corresponding text passages accordingly.

1.2) Figure 5B is also n=2 westerns with no quantification/densitometry analysis and statistical support (p-values). Further quantification and potentially repeats are needed.

The corresponding quantified biological response (measured by stimulation of receptor phosphorylation) towards the proteins has now been added to the figure (Figure 5C) and the quantification shows clear potency differences, which is what the reader is most likely interested in. Figure 5B shows metabolically labelled VEGF-C, which was pulled down by soluble VEGF receptor/IgGFc fusion proteins. Since the soluble VEGF receptors/IgGFc fusion proteins are pre-dimerized via their Fc-tag the affinities does not reflect binding affinities to native receptors (avidity effects, sterical effects, etc.). Hence, our goal with this experiment was only qualitative (binding vs. non-binding). Since we did equalize the protein amounts, we could quantify the binding. However, the difference between the ADAMTS3 and KLK3 forms of VEGF-C becomes apparent only in the – more important – biological activity assay.

1.3) Figure 6 (n=2) lacks quantification/densitometry and p-values. Further quantification and potentially repeats are needed.

We have repeated the Coomassie staining for cathepsin D cleavage of pro- and mature VEGF-C (Figure 6A) and added the quantification as Figure 6B.

1.4) In Figure 4, it is unclear with the current labelling which lanes contain CCBE1. Furthermore if the labels are correct, the experiment lacks a KLK3 only control and a negative control (no KLK3 and no CCBE1). Without the full panel of controls and careful quantification (densitometry analysis and statistical support (p-values)) it is not possible to know if processing is additive or enhanced and the claims are not fully supported by the data.

We have added the densitometric analysis and show the statistical significances now as additional panel B (Figure 4). We have corrected the figure to show which lanes contain CCBE1 (the image got cut off omitting the information about CCBE1). It should be now clear that the figure does contain the important no KLK3/no CCBE1 control. We also show now the results for an experiment where we included the KLK3-only and the CCBE1-only control. The VEGF-C antibody used for this blot does not cross-react with KLK3 or CCBE1 at the concentrations used, and since these blots show co-incubations of purified proteins, the lanes contains only KLK3 or only CCBE1 and appear thus empty.

2) The study provides exclusively evidence for a biochemical capability of KLK3 in VEGF-C regulation and thus is limited in scope. The biological, cellular and tissue level outcomes, consequences and phenotypes associated with the cleavage of VEGFC by KLK3 are not presented in vitro or in vivo. Further information would substantially increase the interest in the study. Does the absence of KLK3 cleavage impact sperm formation or function? Or exogenous administration of KLK3 cleaved vs other forms of VEGF-C differentially impact sperm biology? Is there any evidence that the loss of VEGFC in fact impacts on the formation, maturation or function of sperm?

The biochemical assays provide some evidence about the relative activity of KLK3-cleaved VEGF-C towards VEGFR-3 and VEGFR-2(Figure 1C and D; Figure 5C and D). In order to find support for an in-vivo function for the novel KLK3-form of VEGF-C, we generated Adeno-Associated Viruses producing this form from a truncated cDNA. We performed three injections with this virus into three different locations: skeletal muscle, uterus muscle, oviduct (with increasing relevance for reproductive biology). However, while we have experience with the skeletal muscle model, the other injection sites were a first try. The skeletal muscle showed a lymphangiogenic response to the KLK3-form of VEGF-C. However, the injections into the uterus muscle were not successful and also in the oviduct, we could not see any productive AAV infection (judging from the co-injection of the GFP-control virus). Therefore, we can only conclude that the KLK3-form is active in-vivo, and most likely should influence VEGF-C activity related to lymphangiogenesis in the reproductive system. However, an appropriate model to show the direct effect on reproductive functions is not established yet.

We also performed preliminary studies exposing sperm cells to VEGF-C with very heterogeneous results. Sometimes sperm cells responded to VEGF-C with increased motility, whereas the reaction was weak or completely absent at other times. After ejaculation, activated VEGF-C should immediately start to interact with VEGFRs on the sperm cell membrane if such receptors are present. We speculate that the extent of this activation might determine whether we see a response or not. While it is clear that sperm cells cannot replace receptors by new synthesis, not much is known about receptor recycling in sperm cells. Absent efficient recycling, VEGF receptors on sperm cells can only be activated once. Since it is necessary to liquefy and purify the sperm in order to do motility assays, the window of sensitivity has perhaps passed, and only unusually low endogenous VEGF-C amounts might allow a detectable response by the left-over receptors.

We also did fluorescence microscopy and detected VEGFR-2, VEGFR-3 and VEGF-C on the cell surface of sperm cells. However, due to the extreme density of cell surface proteins on sperm cells, immunohistochemistry is a notoriously unreliable method for expression analysis and – similar to our preliminary sperm motility data – we do not want to include this data yet in this publication and thus risk to report false-positive data.

3) The authors discuss the possibility that KLK3 could be relevant in a tumour setting, which would greatly enhance the scope and interest in the study, however they fail to investigate if this is likely. The manuscript would greatly benefit from an analysis of the distribution of KLK3 in other tissues, especially in pathological tissues and tumours. Analysis of KLK3 distribution in pathological and normal tissues relative to VEGFC distribution would greatly expand the relevance of these findings and interest to researchers.

We agree that expression analysis is very important in this context and we reference now the extensive data, that is already available. A more detailed study than what has been published is clearly warranted but not within the scope of this revision, and we leave this to a forthcoming study of ours (and others). It would be possible to integrate the freely available extensive data from http://ist.medisapiens.com into the supplement (i.e. to assemble the relevant graphical output of for KLK3 and VEGF-C into a supplementary figure). However, linking does perhaps more justice to this continuously updated expression database.

4) Given that KLK3 can activate VEGF-D, is VEGF-D present in seminal fluid and is the cleavage of VEGF-D by KLK3 enhanced by CCBE1?

We performed pull-down of VEGF-D from seminal fluid using

VEGFR-2/Fc and straight loading, but we did not detect VEGF-D in seminal fluid within the detection limit of the anti-VEGF-D antibody (R&D Systems AF286). We mention now in the manuscript in the Results section "Human seminal fluid contains VEGF-C", that we did not detect VEGF-D in seminal plasma and give the corresponding data as supplementary figure. However, we only used one antibody and we cannot exclude that VEGF-D is present in low amounts as we are limited to the sensitivity of the antibody that we used (which is only around 10ng/ml).

The study on the mechanism of VEGF-C/VEGF-D activation by Bui et al., 2016 suggests that VEGF-D activity might be generally independent of CCBE1, and this is one of the reasons why we did not focus our experiments on this question. Despite an additional purification run of CCBE1 this February, we had to focus on VEGF-C due to the limited yield of active CCBE1. The experiments with VEGF-D require a significant amount of CCBE1 as the appropriate protein ratios need to be titrated for each assay since adding too much enzyme does obscure the result (see Jeltsch et al., 2014; Jha et al., 2017). CCBE1 production and purification are still challenging despite reports to the contrary which fail to show that the protein is active with respect to VEGF-C cleavage enhancement.

5) It would be interesting in the context of the paper to know if, similar to cleavage by KLK3, whether cleavage of VEGF-C and VEGF-D by Cathepsin D is enhanced by CCBE1?

We have performed the cleavage of VEGF-C by Cathepsin D plus/minus CCBE1 (two independent experiments). The results indicate that CCBE1 does indeed affect Cathepsin D-cleavage. However, this was only clear in one of the experiments. Hence we need to repeat the experiment a few times before we draw any final conclusions and we regard it as premature to include them in this manuscript at this moment. The same is true for any possible enhancement of VEGF-D cleavage by CCBE1. These experiments require also additional extensive antibody testing since we do not know of an anti-VEGF-D antibody that rivals the sensitivity of the antibodies that we have been using against VEGF-C (882 and sc-374628). Using much higher protein amounts might by itself distort the results in experiments with purified proteins due to stabilization and adsorption effects.

6) The authors show that the different N-teminal cleaved forms of VEGFC have altered binding to VEGFR2 or VEGFR3. This is done at the level of pull-down with Fc forms of R2 and R3. Does this correlate with altered signalling at these receptors for the different ligand forms? Can the authors compare the bioactivity of the different cleavage products shown in Figure 2 at the level of phosphorylation or cell signalling/biology?

We now show the phosphorylation of the receptors as a response to the different forms of VEGF-C in Figure 5C and D. The receptor activation data mirrors the binding data. With progressive shortening of the N-terminal helix, the potential for receptor activation decreases. We have to thank the reviewer since the experiment yielded surprisingly big differences, which answers the questions of many colleagues wondering whether there are any differences in the biological activity of the two major forms of VEGF-C (the plasmin- and the ADAMTS3-generated forms). According to these experiments the differences in the ability to activate the VEGF receptors between the plasmin-form ("minor" form) and the ADAMTS3-form ("major" form) are substantial and this data is the first of its kind to show this clearly.

7) In Figure 6, the authors state that the cleavage of ΔNΔC-VEGF-C by Cathepsin D leads to "super-activation". They show that the processed form is cleaved into even smaller forms in the presence of Cathepsin D. Although Catherpsin D enhances signalling in Figure 7, the claim further processing of ΔNΔC-VEGF-C is hyperactivation at the level of the ligand seems misleading. It is not backed up by any biochemical data on phosphorylation of VEGFR-3 driven by the specific smaller forms of VEGF-C. Could this additional cleavage not be rendering an active form less active? Can the smaller forms be tested in isolation?

The reviewer is entirely correct. "Super-activation" of ΔNΔC-VEGF-C does render it less active (at least towards VEGFR-2). An even more dramatic case is the "super-activation" of VEGF-D by cathepsin D, which abolishes virtually all activity towards VEGFR-3 (while maintaining activity towards VEGFR-2).

We are now using the word *secondary activation* to describe all proteolytic cleavages at the N-terminal region that happen on top of an already existing activation. However, even this could be misinterpreted (since cathepsin D cleavage of VEGF-D abolishes VEGFR-3 binding and is therefore de facto a *partial inactivation*). We explain now the usage of the term *secondary activation* upon first use to avoid this misunderstanding: “We refer to such cleavage on top of an existing activation in the following as “secondary activation” (irrespectively of the receptor activation ability of the resulting protein species).”

We have tried to separate the multiple mature forms of VEGF-C. However, even by analytical HPLC, they did not resolve sufficiently well. This problem is exacerbated as these molecules are dimers and proteolytic cleavage is unlikely synced between the two subunits. A replacement for isolation (with caveats) is the expression from a truncated cDNA, which we show in Figure 5A and D. This data supports the reviewer's point as it shows clearly less binding and activation for the cathepsin-cleaved VEGF-C analogue (VEGF-C-DMH) to VEGFR-2.